

# Ship-based MAX-DOAS measurements of tropospheric NO2, SO2, and HCHO distribution along the Yangtze River

Qianqian Hong[1,#], Cheng Liu[1,2,3,4,#,*], Ka Lok Chan[5,*], Qihou Hu[1], Zhouqing Xie[1,2,3,4], Haoran Liu[2], Fuqi Si[1], Jianguo Liu[1,3]

[1]Key Lab of Environmental Optics and Technology, Anhui Institute of Optics and Fine Mechanics, Hefei Institutes of Physical Science, Chinese Academy of Sciences, Hefei, 230031, China

[2]School of Earth and Space Sciences, University of Science and Technology of China, Hefei, 230026, China

[3]CAS Center for Excellence in Regional Atmospheric Environment, Institute of Urban Environment, Chinese Academy of Sciences, Xiamen, 361021, China

[4]Anhui Province Key Laboratory of Polar Environment and Global Change, USTC, Hefei, 230026, China

[5]Remote Sensing Technology Institute (IMF), German Aerospace Center (DLR), Oberpfaffenhofen, Germany

[#]This two authors contributed equally

[*]*Correspondence to*: Ka Lok Chan (ka.chan@dlr.de), Cheng Liu (chliu81@ustc.edu.cn)

**Abstract.** In this paper, we present ship-based Multi-Axis Differential Optical Absorption Spectroscopy (MAX-DOAS) measurements of tropospheric trace gases distribution along Yangtze River during winter 2015. The measurements were performed along Yangtze River between Shanghai and Wuhan covering major industrial areas in eastern China. Tropospheric vertical column densities (VCDs) of nitrogen dioxide ($NO_2$), sulfur dioxide ($SO_2$), and formaldehyde (HCHO) were retrieved using air mass factor calculated by radiative transfer model. Enhanced tropospheric $NO_2$ and $SO_2$ VCDs were detected over downwind areas of industrial zones over Yangtze River. In addition, spatial distributions of atmospheric pollutants are strongly affected by meteorological conditions, i.e., positive correlations were found between concentration of pollutants and wind speed over these areas indicating strong influence of transportation of pollutants from high-emission upwind areas along Yangtze River. Comparison of tropospheric $NO_2$ VCDs between ship-based MAX-DOAS and OMI satellite observations shows good agreement with each other with Pearson correlation coefficient (R) of 0.82. In this study, $NO_2/SO_2$ ratio was used to estimate the relative contributions of industrial sources and vehicle emissions to ambient $NO_2$ levels. Analysis results of $NO_2/SO_2$ ratio shows that higher contribution of industrial $NO_2$ emissions in Jiangsu province, while $NO_2$ levels in Jiangxi and Hubei provinces are mainly related to vehicle emissions. These results indicate that different pollution control strategies should be applied in different provinces. In addition, multiple linear regression analysis of ambient carbon monoxide (CO) and odd oxygen ($O_x$) indicated that the primary emission and secondary formation of HCHO contribute $54.4 \pm 3.7\%$ and $39.3 \pm 4.3\%$ to the ambient HCHO, respectively. The largest contribution from primary emissions in winter suggested that photochemically induced secondary formation of HCHO is reduced due to lower solar



irradiance in winter. Our findings provide an improved understanding of major pollution sources along the eastern part of Yangtze River which are useful for designing specific air pollution control policies.

## 1 Introduction

Nitrogen dioxide ($NO_2$), sulfur dioxide ($SO_2$), and formaldehyde (HCHO) are important atmospheric constituents playing important roles in tropospheric chemistry. Nitrogen oxides ($NO_x$), defined as the sum of nitric oxide (NO) and $NO_2$, is one of the major pollutant in the troposphere, playing a key role in both tropospheric and stratospheric chemistry. It takes part in the catalytic formation of tropospheric ozone ($O_3$), while being a catalyst for the destruction of stratospheric $O_3$ (Crutzen, 1970). Major sources of $NO_x$ are high-temperature combustions (e.g. fossil fuel burning, biomass burning) and natural processes (e.g. soil microbial activity, lightning events (Lee et al., 1997)). $NO_2$ in high concentration is harmful to human health, especially for immune and respiratory systems. In addition, $NO_2$ can lead to the formation of nitrate aerosols, which is an important component of fine suspended particles in the urban environment. Sulfur dioxide ($SO_2$) is the most abundant anthropogenic sulfur containing air pollutant. In urban areas, $SO_2$ is produced mainly through the combustion of sulfur-containing fossil fuels for power generation and domestic heating, which accounts for more than 75% of the total $SO_2$ emissions (Chin et al., 2000). Atmospheric $SO_2$ causes similar environmental problems as $NO_2$, such as acidification of the natural aqua system, formation of secondary aerosols and causing negative impacts on human health (Chiang et al., 2016). The atmospheric lifetime of both $NO_2$ and $SO_2$ are relatively short, ranging from few hours up to few days (Krotkov et al., 2016), therefore, their spatial distributions are highly influenced by the emission sources.

Formaldehyde (HCHO) is one of most abundant volatile organic compounds (VOCs) in the atmosphere, playing an important role in air quality and atmospheric photochemistry. Incomplete combustion processes including industrial emissions and vehicle exhaust have been identified as the major HCHO sources in the urban atmosphere (Garcia et al., 2006). Formaldehyde can also be produced from the atmospheric photochemical oxidation of methane ($CH_4$) and non-methane hydrocarbons (NMHCs) (Miller et al., 2008). In the polluted regions, terminal alkenes such as isoprene, ethene and propene are the most important HCHO precursors (Goldan et al., 2000). The major sinks of HCHO are photolysis, reaction with OH radical and wet deposition in the atmosphere (Lei et al., 2009). The short atmospheric lifetime of HCHO under sunlight is typically very short (2-4 h), indicating that the daytime ambient HCHO are mostly produced locally (Arlander et al., 1995).

Multi-Axis Differential Optical Absorption Spectroscopy (MAX-DOAS) is a passive remote sensing technique providing indispensable information of atmospheric aerosols and trace gases (Platt and Stutz, 2008). Information of tropospheric trace gases are obtained from the molecular absorption in the ultraviolet and visible wavelength bands by applying the differential optical absorption spectroscopy (DOAS) method to the observations of scattered sun light spectra in several different viewing directions. This method has been widely used for atmospheric $NO_2$, $SO_2$, and HCHO in the past decades (Lee et al., 1997;Heckel et al., 2005;Wang et al., 2014;Hendrick et al., 2014;Chan et al., 2015). MAX-DOAS observations are not only limited to ground based application, but also can be performed on different mobile platforms like



cars (Johansson et al., 2009;Ibrahim et al., 2010;Shaiganfar et al., 2011), aircrafts (Baidar et al., 2013;Dix et al., 2016) or ships (Sinreich et al., 2010;Peters et al., 2012;Takashima et al., 2012;Schreier et al., 2015). In this study, ship-based MAX-

DOAS measurements of $NO_2$, $SO_2$, and HCHO were performed along the Yangtze River. Previous ship-based MAX-DOAS measurements were mainly focused on remote and coastal marine environments to obtain boundary layer background concentrations of trace gases, such as in the Indian and Pacific Ocean. In this study, we performed ship-based MAX-DOAS observations along the Yangtze River, the busiest navigable inland waterway in the world to obtain insight spatial distribution information of trace gases in eastern China.

Yangtze River delta (YRD) is one of the most populated regions in China. Due to the rapid industrialization and urbanization in the past two decades, Yangtze River delta (YRD) is facing a series of air pollution problems. As emission sources are not well characterized and the atmospheric processes are rather complex, it is important to measure the spatial distribution of atmospheric pollutants, i.e., $NO_2$, $SO_2$, and HCHO, in YRD for the investigation of emission sources and atmospheric processes and provides scientific supports for the prevention and designing control measures of air pollution. In

this study, spatial distribution of tropospheric $NO_2$, $SO_2$, and HCHO were retrieved from ship-based MAX-DOAS observations along the Yangtze River between Shanghai and Wuhan. The experiment aims to provide improved understanding of emission sources and atmospheric processes over eastern China, which is potentially useful for the formulation of strategic air pollution control and identification of the effectiveness of air pollution control policies.

In this paper, we present ship-based Multi-Axis Differential Optical Absorption Spectroscopy (MAX-DOAS)

measurements of tropospheric trace gases distribution along the Yangtze River during winter 2015. The measurements were performed along the Yangtze River between Shanghai and Wuhan covering the major industrial areas in eastern China. Details of the experimental setup, spectral analysis and trace gases retrieval of the ship-based MAX-DOAS measurement is presented in section 2. Comparison with OMI $NO_2$ and the contribution of major emission sources to $NO_2$ levels along the Yangtze River, as well as the contribution of primary and secondary sources of HCHO are shown in section 3.

**2 Methodology**

**2.1 The Yangtze River measurement campaign**

The Yangtze River campaign took place in winter 2015 along the Yangtze River over eastern China within the framework of the "Regional Transport and Transformation of Air Pollution in Eastern China". The aims of Yangtze River campaign are to provide better understanding of the transportation and transformation of atmospheric pollution, and to

identify the potential impacts on air quality and climate.

Ship-based measurement campaign was carried out along the eastern part of Yangtze River between Shanghai and Wuhan. The campaign includes a departing journey (from Shanghai to Wuhan) and a returning journey (from Wuhan to Shanghai). The measurement campaign started on 21 November 2015 at 19:30 LT from Shanghai (31.36°N, 121.62°E), a major industrial and commercial hub of eastern China, and arrived in Wuhan (30.62°N, 114.32°E) on 29 November 2015 at



15:42 LT. The measurement ship was then directly sailing back and finally arrived in Shanghai on 4 December 2015 at 20:30 LT. The journey covered most of the major industrial areas in eastern China including several population dense metropolitan cities, such as Nanjing, Wuhu, and Jiujiang. Detail of the cruise track is shown in Fig. 1.

The ship-based MAX-DOAS instrument was part of the air quality monitoring framework of the Yangtze River measurement campaign. The instrument was installed at the beginning of the measurement campaign and started to provide

atmospheric observations on 22 November 2015. A summary of the meteorological conditions during the Yangtze River campaign are shown in Table 1.

### 2.2 MAX-DOAS measurements

### 2.2.1 Experimental setup

Ship-based MAX-DOAS measurements were performed during Yangtze River campaign. The ship-based MAX-DOAS

instrument was developed at the Anhui Institute of Optics and Fine Mechanics (AIOFM), Chinese Academy of Sciences (CAS) which consists of a telescope, a spectrometer and a computer acting as controlling and data acquisition unit. Viewing directions of the telescope are controlled by a stepping motor, scattered sunlight collected by the telescope is redirected to the spectrometer for spectral analysis through an optical fiber bundle. The field of view of the telescope is estimated to be less than 1°. An imaging spectrometer (Princeton instrument), equipped with a charge-coupled device (CCD) detector (512 ×

2048 pixels) is used to measure spectra in the ultraviolet (UV) wavelength range from 303 nm to 370 nm with a spectral resolution of 0.35 nm full width half maximum (FWHM). Spectral data recorded by the imaging spectrometer were average along the first dimension of the CCD in order to get a better signal to noise ratio. During the measurement campaign, the viewing azimuth angle of the telescope was adjusted to 90° (right) relative to the heading direction of the ship (see Fig. S1). A full measurement sequence includes elevation angles ($\alpha$) of 30° and 90° (zenith). The exposure time of each measurement

is set to 100 ms.

### 2.2.2 Data processing and filtering

Although the instrument was positioned in front of the exhaust stack (see Fig. S1), the MAX-DOAS measurements could still be influenced by the exhaust from the ship. Therefore, measurement data contaminated by the ship exhaust were filtered out in our analysis. Individual measurements taken under unfavorable wind directions (relative wind directions

between 150° and 270° with respect to the heading of the ship) were discarded in the following analysis. Due to the stronger absorptions of stratospheric species and low signal to noise ratio at large SZAs, only measurements with solar zenith angle smaller than 75° were taken into account for the DSCDs retrieval.

As the viewing elevation angles of the measurements were relatively high (30° and 90°), therefore, they are insensitivity to the instability or the movement of the ship. In addition, the exposure time of a measurement is rather short (100 ms), the



change of measurement elevation and azimuth angle during one measurement is negligible.

### 2.2.3 DOAS retrieval

Differential slant column densities (DSCDs) of trace gases are derived from the measurement spectra by applying the differential optical absorption spectroscopy (DOAS) technique (Platt and Stutz, 2008). In this study, MAX-DOAS spectra are analyzed using the QDOAS spectral fitting software suite developed by BIRA-IASB (http://uv-
vis.aeronomie.be/software/QDOAS/). The wavelength calibration was performed by using a high resolution solar reference spectrum (Chance and Kurucz, 2010). Dark current (DC) spectrum was taken with exposure time of 3000 ms and number of scan of 20 scans while electronic offset spectrum (OFFSET) was taken with exposure time of 3 ms and number of scan of 20000 scans. The dark current and offset spectra were used to correct measurement spectra prior to the spectra analysis. Several trace gas absorption cross sections (Vandaele et al., 1998;Vandaele et al., 2009;Meller and Moortgat,
2000;Serdyuchenko et al., 2014;Thalman and Volkamer, 2013;Fleischmann et al., 2004), the Ring spectrum, a Frauenhofer reference spectrum and a low order polynomial are included in the DOAS fit. Details of the DOAS fit settings are shown in Table 2. In this study, zenith measurement spectrum with the lowest pollutant concentration was selected as the Frauenhofer reference spectrum for the retrieval of measurement spectra taken during the measurement campaign. Similar reference spectrum selection approaches have been used in different mobile measurements studies (Wu et al., 2013;Li et al., 2015).

Figure 2 shows an example of the DOAS analysis of a spectrum recorded on 1 December 2015 at 14:02 LT with an elevation of 30°. The retrieved $NO_2$ (Fig. 2a), $SO_2$ (Fig. 2b), and HCHO (Fig. 2c) DSCDs are $1.40 \times 10^{17}$, $6.25 \times 10^{16}$, and $3.51 \times 10^{16}$ molec cm$^{-2}$, respectively. In this study, only data with root mean square (RMS) of residuals smaller than $3.0 \times 10^{-3}$ are considered. This filtering criterion (RMS) removed 18.7%, 20.2%, and 25.2% of data for $NO_2$, $SO_2$, and HCHO, respectively.

### 2.2.4 Determination of the tropospheric VCD

The DOAS spectral retrieval results are the differential slant column densities (DSCDs) which are defined as the difference between the slant column density (SCD) of the measured spectrum and the Frauenhofer reference spectrum:

$$DSCD_{meas} = SCD_{meas} - SCD_{Fraunhofer} \tag{1}$$

The SCD is the integrated trace gas concentration along the light path through the atmosphere, which includes both
tropospheric and stratospheric part:

$$SCD_{meas} = SCD_{trop} + SCD_{strat} \tag{2}$$

As scattering of photons most likely takes place in the troposphere, the light path in the stratosphere for zenith and off zenith measurements are very similar. We can assume that $SCD_{strat}(\alpha) \approx SCD_{strat}(90°)$. Then Eq. (1) can be written as:



$$DSCD_{meas}(\alpha) = (SCD_{trop}(\alpha) + SCD_{strat}(\alpha)) - (SCD_{trop}(90°) + SCD_{strat}(90°))$$
$$= SCD_{trop}(\alpha) - SCD_{trop}(90°) \quad\quad\quad\quad\quad\quad\quad (3)$$
$$= DSCD_{trop}(\alpha)$$

where $DSCD_{trop}(\alpha)$ represents the tropospheric DSCD measured at elevation angle of $\alpha$.

As the measured DSCDs are dependent on the absorption path in the atmosphere, the measurement has to convert to vertical column density (VCD) in order to compare with each other. For this purpose, a concept so-called air mass factor (AMF) is applied (Solomon et al., 1987).

$$VCD = \frac{SCD}{AMF} \quad\quad\quad\quad\quad\quad\quad (4)$$

Assuming scattering happens above the trace gas layer, the AMF for the zenith and the off-axis view can be estimated as 1 and $1/\sin\alpha$, respectively (Hönninger et al., 2004). This method is so-called "geometric approximation". However, geometric approximation of AMF could result in large errors under high aerosol load conditions (Wagner et al., 2007). In addition, relative azimuth angle, defined as the angle between the viewing direction and the sun, also plays an important role in the AMF calculation. This effect is particularly important for mobile observations (Wagner et al., 2010). For this reason, we
adapted the simultaneous lidar measurement of aerosol profiles for the radiative transfer calculation of AMFs.

In this study, AMFs for SCD to VCD conversion were calculated using the radiative transfer model SCIATRAN 2.2 (Rozanov et al., 2005). Compared to the geometric approach, radiative transfer calculation of AMF is more computational expensive. Vertical distribution profiles of aerosols and trace gases are also important for the AMF calculation. In this study, trace gas profiles (e.g., $O_3$, $NO_2$, $SO_2$, and HCHO) and vertical profiles of pressure and temperature are taken from WRF-
Chem simulations for AMF calculations (Liu et al., 2016). Aerosol extinction coefficients in the lowest 2 km of the troposphere were taken from the Mie lidar measurements while aerosols above 2 km were not considered in the radiative transfer simulations. Tropospheric AMFs of $NO_2$, $SO_2$, and HCHO were calculated at the central wavelength of their DOAS fitting windows which are 354 nm, 311 nm, and 347 nm, respectively. The aerosol extinction profiles obtained from the Mie lidar are converted to MAX-DOAS retrieval wavelengths assuming a fix Ångström coefficient (Ångström, 1929) of 1. The
aerosol extinction profiles at 354 nm, 311 nm, and 347 nm can be derived using the following formula:

$$\alpha(\lambda_x, z) = \alpha(\lambda_{532}, z) \times (\frac{\lambda_x}{\lambda_{532}})^{-\upsilon} \quad\quad\quad\quad\quad\quad (5)$$

where $\alpha(\lambda_x, z)$ is the aerosol extinction coefficient at wavelength $\lambda_x$; $\alpha(\lambda_{532}, z)$ is the aerosol extinction coefficient at 532 nm; $\upsilon$ is the Ångström coefficient which is assigned to a fix value ($\upsilon=1$) in this study.

A fix set of single scattering albedo (SSA) of 0.95, asymmetry parameter of 0.68 and surface albedo of 0.06 is assumed in
the radiative transfer calculations. In this study, all radiative transfer calculations were performed by using the radiative transfer model SCIATRAN 2.2 (Rozanov et al., 2005). Previous studies show that the uncertainties caused by aerosol single scattering albedo (SSA), aerosol asymmetry parameter (AP) and surface albedo assumptions are less than 10% (Chen et al., 2009;Wang et al., 2012b). Uncertainty of lidar measurement of aerosol extinction profiles also contributes to the uncertainty



in the AMF calculations. A sensitivity study was performed with aerosol profiles with different AODs (i.e., 0.4, 0.6, 0.8, 1.0,
and 1.2) and a single trace gas profile with a constant $NO_2$, $SO_2$, and HCHO concentration of $5.4 \times 10^{11}$ molecules cm$^{-3}$
(equal to 20 ppb at the ground level) to quantify the uncertainty caused by aerosol profiles used in the AMF calculation at
different wavelengths. In the sensitivity analysis, aerosols and trace gases are assumed well mixed in the lowest 0.8 km,
following an exponential decrease with height. The result shows that the variation of AMFs with different aerosol profiles
(SZAs smaller than 75°) are 11%, 13%, and 11% for $NO_2$, $SO_2$, and HCHO, respectively (see Fig. S2). Considering the
uncertainties caused by the assumptions of SSA, AP and surface albedo and uncertainties of aerosol load in the radiative
transfer calculations, we estimated the uncertainties of tropospheric AMFs are ranging between 30-43% for SZAs smaller
than 75°.

For the analysis of MAX-DOAS observations we are mainly focused on the tropospheric vertical column density
($VCD_{trop}$), and the tropospheric quantities can be expressed as follows:

$$VCD_{trop} = \frac{SCD_{trop}(\alpha)}{AMF_{trop}(\alpha)}$$   (6)

As the DOAS analysis results are DSCDs, we have to apply the concept of differential air mass factor (DAMF) to
convert the measurement to vertical columns as follows:

$$
\begin{aligned}
DSCD_{trop}(\alpha) &= SCD_{trop}(\alpha) - SCD_{trop}(90°) \\
&= VCD_{trop} \times AMF_{trop}(\alpha) - VCD_{trop} \times AMF_{trop}(90°) \\
\Rightarrow VCD_{trop} &= \frac{DSCD_{trop}(\alpha)}{AMF_{trop}(\alpha) - AMF_{trop}(90°)} = \frac{DSCD_{trop}(\alpha)}{DAMF_{trop}(\alpha)}
\end{aligned}
$$   (7)

where DAMF is defined as the difference of air mass factor (AMF) between $\alpha \neq 90°$ and $\alpha = 90°$
( $DAMF_{trop}(\alpha) = AMF_{trop}(\alpha) - AMF_{trop}(90°)$ ). This equation (Eq. 7) is regard as the standard method for the determination of
the tropospheric trace gas VCDs from MAX-DOAS observations.

Mobile MAX-DOAS observations are strongly influenced by rapid change of air masses and radiative transfer
conditions along the navigating route. The standard method (Eqs. 1 to 7) to calculate tropospheric VCDs can result in large
errors. An alternative method has been suggested for mobile MAX-DOAS measurements (Wagner et al., 2010). This method
has been applied in previous mobile MAX-DOAS observations (Ibrahim et al., 2010;Wu et al., 2015) and reported to be
better than the standard method for mobile platforms. Therefore, we adapted the new method in this study for tropospheric
VCDs conversion. The tropospheric vertical column density ($VCD_{trop}$) can be expressed as follows (combining Eqs. 1 and 6):

$$
\begin{aligned}
VCD_{trop} &= \frac{SCD_{meas}(\alpha) - SCD_{strat}(SZA)}{AMF_{trop}(\alpha)} \\
&= \frac{DSCD_{meas}(\alpha) + SCD_{ref} - SCD_{strat}(SZA)}{AMF_{trop}(\alpha)}
\end{aligned}
$$   (8)

where SZA denotes the solar zenith angle. We refer to the difference of the two unknowns $SCD_{ref}$ and $SCD_{strat}(SZA)$ as





DSCD$_{offset}$(SZA) and can be written as follows:

$$DSCD_{offset}\ (SZA) = SCD_{ref} - SCD_{strat}\ (SZA) \tag{9}$$

The expressions for VCD$_{trop}$ in Eqs. 7 and 8 are set equal:

$$\frac{DSCD_{meas}(\alpha) - DSCD_{meas}(90°)}{AMF_{trop}(\alpha) - AMF_{trop}(90°)}$$
$$= \frac{DSCD_{meas}(\alpha) + SCD_{ref} - SCD_{strat}(SZA)}{AMF_{trop}(\alpha)} \tag{10}$$

This equation can be solved for DSCD$_{offset}$(SZA) as defined in Eq. 9:

$$DSCD_{offset}\ (SZA)$$
$$= \frac{AMF_{trop}(90°) \times DSCD_{meas}(\alpha) - AMF_{trop}(\alpha) \times DSCD_{meas}(90°)}{AMF_{trop}(\alpha) - AMF_{trop}(90°)} \tag{11}$$

Since DSCD$_{offset}$(SZA) is a smooth function of the SZA or time, we can fit the time series of calculated DSCD$_{offset}$(SZA or t$_i$) by a low order polynomial (second order). t$_i$ indicates the time between the two selected measurements from one elevation sequence i, the time series of the calculated DSCD$_{offset}$(SZA) can be written as:

$$DSCD_{offset}\ (t_i)$$
$$= \frac{AMF_{trop}(90°, t_i) \times DSCD_{meas}(\alpha, t_i) - AMF_{trop}(\alpha, t_i) \times DSCD_{meas}(90°, t_i)}{AMF_{trop}(\alpha, t_i) - AMF_{trop}(90°, t_i)} \tag{12}$$

The fitted polynomial represents the approximation of DSCD$_{offset}$(t$_i$) and can be inserted into Eq. 8. In this way we can obtain a time series of tropospheric trace gas VCDs essentially without errors introduced by the spatio-temporal variations of the trace gas field. The detailed description of this new method can be found in Wagner et al. (2010).

### 2.3 OMI Satellite observations

     The Ozone Monitoring Instrument (OMI) was launched onboard the NASA Earth Observing System (EOS)-Aura
satellite on 15 July 2004 (Levelt et al., 2006). It is a nadir-viewing imaging spectrometer measuring direct and Earth's reflected sunlight in the ultra violet (UV) and visible (VIS) range from 270 to 500 nm. OMI aims to monitor global atmospheric trace gases distribution with high spatial (up to $13 \times 24$ km) and temporal (daily global coverage) resolution. The local overpass time of OMI is between 13:40 and 13:50 (local time) on the ascending node. In this study, USTC's OMI tropospheric NO$_2$ product is used (Liu et al., 2016). Slant column densities (SCDs) of NO$_2$ are retrieved by applying the
DOAS fit to OMI spectra. Separation of stratospheric and tropospheric columns is achieved by the local analysis of the stratospheric field over unpolluted areas (Bucsela et al., 2013;Krotkov et al., 2017).The OMI NO$_2$ SCDs are converted to VCDs by using the concept of air mass factor (AMF). The AMFs are calculated based on the NO$_2$ and atmospheric profiles derived from WRF-Chem chemistry transport model simulations with a horizontal resolution of $20 \times 20$ km over eastern China. In this study, the National Centers for Environmental Prediction (NCEP) Final operational global analysis (FNL)



meteorological data are used to drive the WRF-Chem simulations. Details of the chemistry transport model simulation as well as the satellite data retrieval process can be found in Liu et al. (2016).

**2.4 Ancillary data**

Lidar observations of aerosol vertical distribution were also carried out during the campaign. The lidar system is equipped with a diode-pumped frequency doubled Nd:YAG laser emitting laser pulses at 532 nm. The typical pulse energy
of the laser is about 20 mJ with a pulse repetition frequency of 20 Hz. The laser beam is emitted with divergence of 1 mrad and 200 mm off-axis to the receiving telescope with a field of view of 2 mrad, resulting in an overlap height of about 195 m. A constant lidar ratio ($S_p$, extinction to backscatter ratio) of 50 sr was assumed in the lidar retrieval. Details of the lidar system and the data retrieval can be found in (Chen et al., 2017).

Meteorological parameters such as wind direction and wind speed were obtained from an automatic weather station on
board the measurement ship. In-situ trace gas measurements, such as CO, $O_3$, and $NO_2$ were performed using Sensor Networks for Air Quality (SNAQ) during the campaign. SNAQ is a highly portable and low-cost air quality measurement network methodology incorporating electrochemical gas sensors which can be used for high resolution air quality studies at ppbv levels (Mead et al., 2013). CO, $O_3$, and $NO_2$ were monitored by SNAQ with a 20s resolution, and the detection limit of CO, $O_3$, and $NO_2$ were 3 ppbv+5% of measured CO, 2 ppbv+5% of measured $O_3$, and 2 ppbv+5% of measured $NO_2$,
respectively.

**3 Results and discussion**

MAX-DOAS measurements were conducted during the campaign period from 22 November to 4 December 2015. The measurements were interrupted occasionally due to power failure of the measurement ship and instrumental problems, details of the measurement period are listed in Table 1. All the times reported herein are local time (LT=UTC+8).
Measurement spectra taken during daytime between 08:00 and 16:00 (SZAs smaller than 75°), corresponding to the sunshine period during wintertime in China were used for analysis. In order to avoid unnecessary uncertainties introduced during the VCD conversion, we use the radiative transfer model with lidar aerosol profiles as input for the AMF calculation to convert all the measurements to VCDs.

**3.1 General characteristics of tropospheric $NO_2$, $SO_2$, and HCHO**

Time series of tropospheric $NO_2$, $SO_2$, and HCHO vertical column densities (VCDs) for the entire campaign from 22 November to 4 December, 2015 are shown in Figs. 3-5. Missing data are due to power failure of the measurement ship and instrumental problems, measurements taken under unfavorable wind directions and SZAs larger than 75°. The mean $NO_2$



VCD of the entire campaign is $2.27 \times 10^{16}$ molec cm$^{-2}$, with an exceptionally large variation range from $1.31 \times 10^{15}$ molec cm$^{-2}$ to $7.72 \times 10^{16}$ molec cm$^{-2}$. About half of the NO$_2$ VCDs are in the range of 5-20 ($\times 10^{15}$ molec cm$^{-2}$) and high NO$_2$

VCDs (i.e., $> 5 \times 10^{16}$ molec cm$^{-2}$, about the 95th percentile value) are about 2.2 times higher than mean value (Fig. S3a). The mean SO$_2$ VCD of the entire campaign is $2.14 \times 10^{16}$ molec cm$^{-2}$ with a range from $1.05 \times 10^{15}$ molec cm$^{-2}$ to $9.29 \times 10^{16}$ molec cm$^{-2}$. Although more than half of the values are in the range of 10-25 ($\times 10^{15}$ molec cm$^{-2}$), high SO$_2$ VCDs (i.e., $> 5.5 \times 10^{16}$ molec cm$^{-2}$, about the 95th percentile value) are about 2.5 times higher than mean value (Fig. S3b). It should be noted that three elevated tropospheric NO$_2$ and SO$_2$ VCDs events have been observed which are highlighted in gray in Figs.

3-4. The mean HCHO VCDs of the entire campaign is $9.61 \times 10^{15}$ molec cm$^{-2}$, ranging from $1.05 \times 10^{15}$ molec cm$^{-2}$ to $5.37 \times 10^{16}$ molec cm$^{-2}$. Most of the HCHO VCDs lie between 1-14 ($\times 10^{15}$ molec cm$^{-2}$) and high HCHO VCDs (i.e., $>2.6 \times 10^{16}$ molec cm$^{-2}$, about the 95th percentile value) are about 2.7 times higher than mean value (see Fig. S3c).

The spatial distribution of NO$_2$, SO$_2$, and HCHO VCDs along the eastern part of the Yangtze River are shown in Fig. 6. Three elevated tropospheric NO$_2$ and SO$_2$ VCDs events over Yangtze River were detected around three major industrial

cities with large number of heavy emission sources, i.e., Jiujiang (#1), an industrial city located on the southern shores of the Yangtze River in northwest Jiangxi Province, China; Nanjing (#2), the provincial capital and the most populous city in Jiangsu province, Eastern China; and Shanghai (#3), a metropolis located in the Yangtze River Delta with the busiest container port in the world.

The variations of the NO$_2$ and SO$_2$ VCDs are closely linked to the spatial distribution of emission sources around

industrial cities as well as meteorological conditions (e.g., wind speed and wind direction). Most of the previous studies show an inverse relationship between wind speed and air quality, i.e., the lower the wind speed the higher the pollution level, which suggesting that low wind speed condition is limiting the mixing and dispersion of atmospheric pollutants, and thus in favors of accumulation of local emissions (Chan et al., 2012;Wang et al., 2012b;Chan et al., 2014;Chan et al., 2017;Wang et al., 2017). However, positive correlations ($p < 0.05$, t-test) were found between the mobile MAX-DOAS NO$_2$ and SO$_2$ VCDs

(red circle and blue square, respectively) and wind speed during these three events (Fig. 7a-7c). The result suggested that these episodes are most likely not related to accumulation of local emission. We further investigated the possible influence of transport on the NO$_2$ spatial distribution by looking into the backward trajectories during these episodes. We calculated 24h backward trajectories of air masses using the HYSPLIT (Hybrid Single-Particle Lagrangian Integrated Trajectory) model which is developed by the National Oceanic and Atmospheric Administration-Air Resource Laboratory (NOAA-ARL)

(Stein et al., 2015) (http://ready.arl.noaa.gov/HYSPLIT.php). Meteorological data from the Global Data Assimilation System (GDAS) with a spatial resolution of 1°×1° and 24 vertical levels was used in the model for the trajectory simulations. The 24h backward trajectories (green marks indicate starting point, −6, −12, −18 and −24 h, respectively) calculated by the NOAA HYSPLIT model is shown in Fig. 7d for all three episodes. As indicated by the 24h back-trajectories, NO$_2$ VCDs on 1 December 2015 (Event #1) are prominent when under southwesterly wind condition. This is probably due to an industrial

city (Jiujiang) is located on the southern shores of the Yangtze River at the upwind areas of the ship. Backward trajectories (Events #2 and #3, Fig. 7d) indicated prevailing northwesterly wind during 3-4 December 2015. In addition, the backward





trajectory analysis suggested that rapid transport of air masses carries significant amount of pollutants from polluted areas in northern China (Krotkov et al., 2016) across the Yangtze River, resulting in higher $NO_2$ and $SO_2$ VCDs. The higher the wind speed the higher the $NO_2$ and $SO_2$ VCDs under northwesterly wind conditions (Figs. 7b and 7c), which means that the

transport from distant sources is more significant than of the contribution from local emission sources. These result suggested that the spatial distributions of pollutants along the Yangtze River are strongly influenced by the meteorological condition. In comparison to these high VCDs events, relatively low $NO_2$ and $SO_2$ VCDs were observed in the first few days of the campaign (22 to 25 November), which might due to occasional showers during these days removed pollutants through wet deposition.

Elevated tropospheric HCHO VCDs (up to $2 \times 10^{16}$ molec $cm^{-2}$) are observed mostly during clear days (e.g., 26 November, 3 and 4 December), with good visibility and low cloud coverage (see Table 1). This is probably due to the enhancement of photochemical formation of atmospheric HCHO under strong solar irradiation. In contrast, lower HCHO VCDs were observed mainly on rainy, cloudy and haze days. Elevated HCHO VCDs in the HCHO time series was found on 3 December 2015 when the ship was anchored at Yizheng Marine department (32.25°N, 119.15°E). This day is mostly

cloud-free with good visibility. In addition, HCHO VCDs show a diurnal pattern with low values in the morning and late afternoon and peaks around noontime. This diurnal pattern indicates the significant contribution of photochemical formation of HCHO. Detailed analysis of the primary sources and secondary formation of HCHO for 3 December 2015 is shown in section 3.4.

### 3.2 Comparison with OMI NO$_2$

In order to compare the ship-based MAX-DOAS measurements to OMI observations, the ship-based MAX-DOAS data are temporally averaged around the OMI satellite overpass time from 12:00 to 14:00 (LT). For 22, 25, and 26 November, no MAX-DOAS data were available during the OMI overpass time due to power failure of the measurement ship and instrumental problems (see Table 1). All OMI measurement within 20 km (ship speed: ~10-20 km/h) radius of the ship's averaged position from 12:00 to 14:00 (LT) are averaged and compared to the averaged ship MAX-DOAS data. For 29

November and 1, 3 December, no satellite observations were available at the corresponding ship's location. As a result, 7 days (Nov 23, 24, 27, 28 and 30 and Dec 1 and 3) of measurements from both OMI and ship-based MAX-DOAS are used for data comparison.

Scatter plot of the ship MAX-DOAS and OMI $NO_2$ measurements is shown in Fig. 8a. Tropospheric $NO_2$ VCDs measured by ship-based MAX-DOAS and OMI shows good agreement with Pearson correlation coefficient (R) of 0.82.

However, the regression analysis indicated that the OMI data underestimated the tropospheric $NO_2$ VCD by about 10%. Time series of ship-based MAX-DOAS and OMI $NO_2$ VCDs is shown in Fig. 8b. The ship-based MAX-DOAS data were higher than the OMI values for most of the time. Underestimation of tropospheric $NO_2$ VCDs for OMI might attribute to the averaging effect over large OMI pixel. Unpolluted or less polluted areas are also included in the OMI pixel and resulting in



low values over pollution hotspots.

In order to have a better insight of the spatial distribution pattern of tropospheric $NO_2$ along the Yangtze River, $NO_2$ VCDs measured by both ship-based MAX-DOAS and OMI are plot on same map which is shown in Fig. 9. OMI tropospheric $NO_2$ VCDs are gridded onto a 0.02° × 0.02° grid using the parabolic spline gridding algorithm (Kuhlmann et al., 2014;Chan et al., 2015;Chan et al., 2017). The gridding routine was reported to provide more realistic continuous spatial distributions of $NO_2$ while preserving the details of emission hotspots (Chan et al., 2015;Chan et al., 2017). During clear

days (27 November and 4 December, 2015), both ship-based MAX-DOAS and OMI capture similar $NO_2$ spatial pattern (Fig. 9a and 9d). However, a significant enhancement of $NO_2$ close to the exit of the Yangtze River was observed by ship-based MAX-DOAS observation, as shown in Fig. 9d which does not show up in the OMI observation. This is probably due to the mismatch of OMI overpass time and the ship-based MAX-DOAS measurement time. The spatial coverage of OMI observation was limited on 30 November 2015 due to cloudy sky condition. On the other hand, $NO_2$ hotspots can be

observed from the ship-based measurement as shown in Fig. 9b. These $NO_2$ peaks cannot be detected by OMI as cloud shield $NO_2$ at the lower troposphere. Different spatial patterns were detected by the ship-based MAX-DOAS and OMI satellite on 2 December 2015 which was a haze day (Fig. 9c). The discrepancy might be due to the strong influence by the aerosols which has a strong influence on the radiative transfer in the atmosphere. The AMFs of ship-based MAX-DOAS data were calculated using lidar aerosol profile as input while the OMI product does not consider aerosol and therefore resulting

in a larger discrepancy under heavy aerosol load conditions.

Ship-based MAX-DOAS is more sensitivity to near surface $NO_2$, however, the measurement is only limited to the area along the Yangtze River. In order to have a broader coverage of tropospheric $NO_2$ distribution over Yangtze River delta, OMI tropospheric $NO_2$ product is used in this study. Figure 10 shows the averaged tropospheric $NO_2$ VCDs along Yangtze River and its surrounding regions (26°-34°N, 112°-124°E). Enhanced $NO_2$ VCDs appeared at the exit of Yangtze River. This

area includes southern Jiangsu, eastern Anhui, and northern Zhejiang. It is obvious that the pollution level along Yangtze River (white line in Fig. 10) is higher than surrounding areas, especially areas from Wuhu to Wuhan. This is probably resulting from the fact that most of the industrial activities are concentrated along Yangtze River as the logistic is much more convenience and cost efficient through water transportation. Our observation implies that specific emission control measures should be applied on the highly polluted industry along Yangtze River.

**3.3 Possible contributions to ambient $NO_2$ levels**

Fossil fuel consumption is the major source of anthropogenic $NO_x$ emissions, especially in highly industrialized and urbanized regions. Industrial sources (including power plants, other fuel combustion facilities, and non-combustion processes) and vehicle emissions are the two major contributors, which together composed about 90% of the total anthropogenic $NO_x$ emissions in China (Huang et al., 2011;Shi et al., 2014). Understanding the individual contributions of

industrial sources and vehicle exhaust to ambient $NO_2$ is important for designing suitable emission control strategies in



polluted areas, like Yangtze River delta. In this study, we estimated contribution of different emission sources to $NO_2$ levels along the Yangtze River by analyzing the ratio between ambient $NO_2$ and $SO_2$.

Figure 1 (green dots) shows a number of power plants are located along the Yangtze River between Shanghai and Wuhan. Industrial zones can also be found close to these power plants along the Yangtze River due to the logistically

convenience. Emissions of $NO_2$ and $SO_2$ from coal-fired power plants are significant air pollution sources. As the atmospheric lifetime of $NO_2$ and $SO_2$ is roughly the same (Krotkov et al., 2016), the ambient $NO_2/SO_2$ ratio is approximately equal to the emission ratio of $NO_2/SO_2$. In addition, vehicles mainly emit $NO_x$ and their $SO_2$ emissions are trivial, while coal-fired power plants, heavy industries and ships mostly use sulfur-containing fossil fuels which emit both $NO_x$ and $SO_2$. Therefore, lower $NO_2/SO_2$ ratio implies larger contribution from industrial sources (Zhang et al., 2017), while higher

$NO_2/SO_2$ ratios indicate larger contribution from vehicle exhaust sources toward $NO_2$ levels (Mallik and Lal, 2014;Krotkov et al., 2016). It should be noted that it is difficult to separate local ship emissions from industrial emissions due to their similar emission components, so industrial sources in this paper including not only coal-fired power plants and heavy industries but also ship emissions. In this study, we analyzed $NO_2/SO_2$ ratios around coal-fired power plants along the Yangtze River. $NO_2/SO_2$ ratios are determined by linear regression of $NO_2$ and $SO_2$ measurements around power plants

which is shown in Fig. 11. $NO_2$ and $SO_2$ VCDs measured within 2 km of the power plants are used for the analysis. Good correlation was found between $NO_2$ and $SO_2$ VCDs measured around coal-fired power plants (R = 0.91, N = 195) which implies the ambient $NO_2$ and $SO_2$ are mostly emitted from similar sources (i.e., coal combustion). The slope of linear regression is $0.56 \pm 0.02$. Relatively low $NO_2/SO_2$ ratio indicates large contributions from combustion of high sulfur containing fuel, e.g. coal, which is mainly used for power generation. In addition, the desulfurization filters installed in these

power plants are either ineffective or maybe even deactivated during the time.

Assuming the ambient $NO_2$ is mostly emitted from industrial sources (mainly from power plants) and vehicle exhaust, other anthropogenic sources like biomass burning and natural sources are negligible. The $NO_2/SO_2$ ratio (slope) and intercept (offset) of the linear regression of data measured around the power plants can be used to estimate the source contributions of industrial sources and vehicle exhaust to the ambient $NO_2$ concentration. Assuming the $NO_2/SO_2$ ratio for industrial emission

is constant, we can estimated the industrial and vehicle contribution by using the following equations:

$$
\begin{aligned}
P_{\text{Industrial sources}} &= \frac{NO_2(\text{power plants})}{NO_2(\text{total})} \times 100\% \\
&= \frac{SO_2 \times \text{Slope} + \text{Intercept}}{NO_2(\text{total})} \times 100\%
\end{aligned}
\tag{13}
$$

$$
\begin{aligned}
P_{\text{vehicle exhaust}} &= \frac{NO_2(\text{vehicle exhaust})}{NO_2(\text{total})} \times 100\% \\
&= \frac{NO_2(\text{total}) - (SO_2 \times \text{Slope} + \text{Intercept})}{NO_2(\text{total})} \times 100\%
\end{aligned}
\tag{14}
$$

where $SO_2$ represents the $SO_2$ VCDs, $NO_2$(total) denotes the $NO_2$ VCDs, the slope and the intercept of the linear regression





are 0.56 and 1.86 (unit: $10^{15}$ molec cm$^{-2}$), respectively.

During the campaign, the route covered four provinces along the Yangtze River, i.e., Jiangsu, Anhui, Jiangxi, and Hubei. Figure 12 shows the relative contributions of industrial sources and vehicle exhaust to the ambient $NO_2$ levels over the four provinces. In Jiangsu province, a higher contribution from power generation to $NO_2$ level was found, which is mainly due to large number of power plant located along the Yangtze River in Jiangsu province. The $NO_x$ emission over the eastern China, including Jiangsu province (15.41 t/(km$^2$·y)) (Shi et al., 2014), is more intensive than other parts of China. In

addition, Jiangsu province is one of the province in China with maximum annual $NO_x$ emissions from coal-fired power plants (Zhao et al., 2008;Wang et al., 2012a). In contrast, the contribution of vehicle exhaust to $NO_2$ level was higher than that of coal-fired power plants in Jiangxi and Hubei provinces, suggesting that traffic emissions have larger impacts on the $NO_2$ level in these provinces. In Anhui province, the contributions of coal-fired power plants and vehicle exhaust to $NO_2$ level were about the same. Our result suggests that different pollution controlling strategy should be applied in different

province: power generation emissions are the major reduction target for Jiangsu province; while more specific control policies are need to reduce the vehicle exhaust pollution in Jiangxi and Hubei provinces.

### 3.4 Estimation of primary and secondary sources of ambient HCHO

Industrial zones are mainly located along the Yangtze River due to the logistically convenience. Observations along the Yangtze River were constantly influenced by plumes originating from various industrial activities, such as coal burning,

crude oil refining and plastic and rubber syntheses. Besides the direct primary emissions, ambient HCHO can also formed through secondary atmospheric processes. Therefore, it is important to quantify the contribution of primary and secondary HCHO in order to better understand the atmospheric processes as well as the corresponding impacts on the local air quality.

CO is directly emitted to the atmosphere through combustion processes (e.g., incomplete combustion of vehicle engines) and therefore can be use as a tracer for primary emission of HCHO (Friedfeld et al., 2002;Garcia et al., 2006). On the other

hand, $O_3$ reacts with NO emitted from automobiles to form $NO_2$. Thus, the odd oxygen $O_x$ ($O_x = O_3 + NO_2$) is often used as a tracer for photochemical processes in urban atmosphere (Wood et al., 2010). In this study, we use CO as the tracer of primary HCHO while $O_x$ being an indicator of secondary HCHO formation. The CO and $O_x$ data were measured by Sensor Networks for Air Quality (SNAQ) during this campaign. Previous study shows that a linear model can be used for the source appointment analysis of ambient HCHO (Garcia et al., 2006). The measured HCHO was apportioned by a multiple linear

regression model which parameterized by the following equation:

$$[\text{HCHO}] = \beta_0 + \beta_1[\text{CO}] + \beta_2[\text{O}_x] \tag{15}$$

where $\beta_0$, $\beta_1$, and $\beta_2$ are the fit coefficients obtained from the multiple linear regression. The analysis was done in daily bases.

The relative contributions of primary emission, photochemical formation and atmospheric background HCHO to the total atmospheric HCHO are calculated according to the tracer concentrations and corresponding fit coefficients by the

following equations:





$$P_{Primary} = \frac{\beta_1 [CO]_i}{\beta_0 + \beta_1 [CO]_i + \beta_2 [O_x]_i} \times 100\% \tag{16}$$

$$P_{Secondary} = \frac{\beta_2 [O_x]_i}{\beta_0 + \beta_1 [CO]_i + \beta_2 [O_x]_i} \times 100\% \tag{17}$$

$$P_{Background} = \frac{\beta_0}{\beta_0 + \beta_1 [CO]_i + \beta_2 [O_x]_i} \times 100\% \tag{18}$$

where $P_{Primary}$ represents the contribution from primary sources (vehicle and industrial emissions); $P_{Secondary}$ is the
contribution of secondary HCHO (photochemical oxidation); and $P_{Background}$ indicates the background HCHO which is neither
classified as primary nor secondary HCHO. According to previous studies in YRD (Wang et al., 2015;Ma et al., 2016), the
background level of HCHO is limited to 1 ppbv. Therefore, the regression parameter $\beta_0$ is fixed at 1 ppbv in this analysis.
$[CO]_i$ and $[O_x]_i$ represent the concentrations of CO and $O_x$ at time i, respectively. $\beta_1$ and $\beta_2$ are the regression coefficients
obtained from multiple linear regressions.

As other factors can also affect the atmospheric HCHO concentration, in order to make sure the regression model is
representative for atmospheric conditions, only data fulfilling the following criteria are used in the analysis (a) correlation
coefficient (R) larger than 0.75 (Li et al., 2010) and (b) significance value lower than 0.05. Of the 12 days of measurements,
only 2 days fulfill the criteria to be considered in this analysis. The parameters of the multiple linear regression fit and the
linear Pearson correlation coefficient (R) for the measured and modeled HCHO are shown in Table 3.

Time series of measured and modeled HCHO of 3 December 2015 is shown in Fig. 13a. Both the measured and
modeled HCHO concentrations show similar temporal development with a rising trend in the morning and reached the peak
value at noon, then followed by a decrease in the afternoon. The linear regression between measured and modeled HCHO
shows a reasonably good agreement with slope of 0.98 and Pearson correlation coefficient (R) of 0.78 on December 3, 2015
(see Fig. 13b). All measurements lie within the 95% prediction interval indicating the best estimate of modeled HCHO. The
regression model could not fully reconstruct the measurements, indicating there are other factors influencing the atmospheric
HCHO levels. Due to the complexity of emission sources, a constant CO/HCHO factor might not be good enough to
represent the HCHO emission from all primary sources. A number of petrochemical-related manufactures and organic
synthesis processes industry located along the Yangtze River resulting in higher HCHO emissions and the CO emission
factor varies with their emission processes. In addition, as we were measuring on a mobile platform, the composition of
emission could change with the measurement location. Future investigation could focus on characterizing the primary
industrial emissions of HCHO by different sector.

The diurnal variation of HCHO contribution from primary sources, secondary formation and background contributions
on 3 December 2015 is shown in Fig. 14. Background HCHO only accounts for a small portion (6.2 ± 0.8%, average ± S.D.)
of the total ambient HCHO, while the primary sources contribute the largest fraction of the ambient HCHO, with an average
percentage of 54.4 ± 3.7%. The primary sources contributions were relatively stable, which might be due to industrial
emission does not show a significant diurnal pattern. The contribution associated with secondary formation accounted for



roughly $39.3 \pm 4.3\%$ of the total HCHO on daily average. Secondary formation of HCHO shows a peak value during noon time (11:00-14:00) which is mainly due to enhancement of photochemical reaction during noon time. Our result is consistence with a similar study in Rome, Italy, which the secondary contribution of HCHO is about 35% during winter time

(Possanzini et al., 2002). While secondary formation of HCHO has been reported as the largest ambient HCHO source in summer (Parrish et al., 2012;Ling et al., 2017). Reduced photochemical reaction in winter resulting in lower formation rate of HCHO, and therefore, primary emissions become the major source of ambient HCHO in winter.

## 4 Summary

In this paper, we present the ship-based MAX-DOAS measurements along Yangtze River from Shanghai to Wuhan
(22 November to 4 December, 2015). Scattered sunlight spectra were measured to retrieve differential tropospheric slant column densities (DSCDs) of $NO_2$, $SO_2$, and HCHO. DSCDs of $NO_2$, $SO_2$, and HCHO were converted to tropospheric vertical column densities (VCDs) using AMF computed by Radiative Transfer Model with lidar aerosol profile as input. During the campaign, three significantly enhanced tropospheric $NO_2$ and $SO_2$ VCDs events were detected over the downwind areas of industrial zones. Spatial distributions of atmospheric pollutants are strongly affected by meteorological
conditions, i.e., positive correlations were found between the ship-based MAX-DOAS data and wind speed for these three events, which indicates that the transportation of pollutants from the high-emission areas have a strong influence on the $NO_2$ and $SO_2$ distribution along Yangtze River. Comparison of tropospheric $NO_2$ VCDs between ship-based MAX-DOAS and OMI satellite observations shows good agreement with Pearson correlation coefficient (R) of 0.82. However, OMI underestimated tropospheric $NO_2$ by 10% which is mainly due to the averaging effect over large OMI pixels. In addition,
satellite observations have lower sensitivity to near-surface pollutants compared to ground based measurements.

In this study, $NO_2/SO_2$ ratio is used to quantify relative contributions of industrial sources and vehicle emissions to ambient $NO_2$ levels. The result shows that Jiangsu province has a higher contribution from industrial sources due to the large number of power plants situated along the Yangtze River in Jiangsu. In contrast, contributions from vehicle emissions to $NO_2$ level are higher than that of industrial sources in Jiangxi and Hubei provinces. Our result suggested that traffic volume
has large impact on $NO_2$ level in these provinces. These results indicate that different $NO_2$ pollution control strategy should be applied in different provinces. In addition, we estimated the contributions of primary and secondary emission sources to ambient HCHO levels using a multiple linear regression method. Result from 3 December 2015 indicated that primary sources have the largest contribution to the ambient HCHO ($54.4 \pm 3.7\%$), while secondary formation contributes $39.3 \pm 4.3\%$ of the total ambient HCHO. The remaining fraction $6.2 \pm 0.8\%$ is attributed to the background. The largest contribution
from primary sources in winter suggested that photochemically induced secondary formation of ambient HCHO is reduced due to lower solar irradiance in winter. This study provides an improved understanding of the impacts of different emission sources in different provinces along the eastern part of Yangtze River to the local air quality. Our findings are useful for designing specific air pollution control and environmental policies.



*Acknowledgement*

The authors would like to thank Prof. Jianmin Chen's group from Fudan University for the organization of the Yangtze River measurement campaign. We would also like to thank the National Oceanic and Atmospheric Administration (NOAA) Air Resources Laboratory (ARL) for the provision of the HYSPLIT transport and dispersion model used in this publication. We thank Fudan University and Cambridge University in providing meteorological measurements and measurements of other atmospheric trace pollutants. The work presented in this paper is jointly supported by the National Key Project of MOST

(project no. 2016YFC0203302, 2016YFC0200404), the National Natural Science Foundation of China (project no. 41575021, 91544212, 41722501, 51778596), Key Project of CAS (project no. KJZD-EW-TZ-G06-01) and the National Key R&D Program of China (project no. 2017YFC0210002).

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





**Table 1.** Weather and viewing conditions during Yangtze River campaign from 22 November to 4 December, 2015.
Interruption of the ship based MAX-DOAS is also listed.

| Period | Conditions | | |
|---|---|---|---|
| 22-25 November | Rain and occasional fog occurred in the morning and night | | |
| 26-27 November | Improving weather and viewing conditions, sunny | | |
| 28 November | Occasional light rain and partly cloudy around noon | | |
| 29 November | Haze | | |
| 30 November | Partly cloudy | | |
| 1-2 December | Haze | | |
| 3-4 December | Good viewing conditions, almost clear sky | | |
| Interruption event | Start time (yyyy/mm/dd, UTC + 8 h) | End time (yyyy/mm/dd, UTC + 8 h) | Reason for interruption |
| 1 | 2015/11/22 11:00 | 2015/11/22 14:25 | power outages |
| 2 | 2015/11/25 7:00 | 2015/11/25 18:00 | frequent power outages all day long |
| 3 | 2015/11/26 11:40 | 2015/11/26 15:00 | instrumental problems |



**Table 2.** Summary of the DOAS retrieval settings used for the $NO_2$, $SO_2$, and HCHO slant column densities retrieval.

| Parameter | Data source | Fitting internal (nm) | | |
|---|---|---|---|---|
| | | $NO_2$ | $SO_2$ | HCHO |
| Wavelength range | | 338-368 nm | 308-314 nm | 336.5-359 nm |
| $NO_2$ | Vandaele et al. (1998), 220K, 294K, $I_0$-correction* (SCD of $10^{17}$ molecules/cm$^2$) | √ | √ (only 294K) | √ (only 294K) |
| $SO_2$ | Vandaele et al. (2009), 298K | × | √ | × |
| HCHO | Meller and Moortgat (2000), 297K | √ | × | √ |
| $O_3$ | Serdyuchenko et al. (2014), 223K, 243K, $I_0$-correction* (SCD of $10^{20}$ molecules/cm$^2$) | √ | √ | √ |
| $O_4$ | Thalman and Volkamer (2013), 293K | √ | × | √ |
| BrO | Fleischmann et al. (2004), 223K | √ | × | √ |
| Ring | Ring spectra calculated with QDOAS according to Chance and Spurr (1997) | √ | √ | √ |
| Polynomial degree | | Order 5 | Order 5 | Order 5 |
| Intensity offset | | Constant | Order 1 | Order 1 |
| Wavelength calibration | Based on a high resolution solar reference spectrum (SAO2010 solar spectra) (Chance and Kurucz, 2010) | | | |

*Solar $I_0$-correction (Aliwell et al., 2002)



**Table 3.** Coefficients of the multiple linear regression and the correlation coefficient (R) for the measured and modeled HCHO in Eq. (15). $\beta_1$ represents the emission ratio of HCHO with respect to CO. $\beta_2$ denotes the portion of HCHO from photochemical production, while $\beta_0$ represents HCHO background concentration which is fixed to 1 ppbv.

| Date (yyyy/mm/dd) | $\beta_1$ | $\beta_2$ | $\beta_0$ | R | N.Obs |
|---|---|---|---|---|---|
| 2015/11/26 | 0.0165±0.0083 | 0.0765±0.0323 | 1.0 | 0.7937 | 80 |
| 2015/12/03 | 0.0312±0.0097 | 0.1149±0.0514 | 1.0 | 0.7746 | 139 |




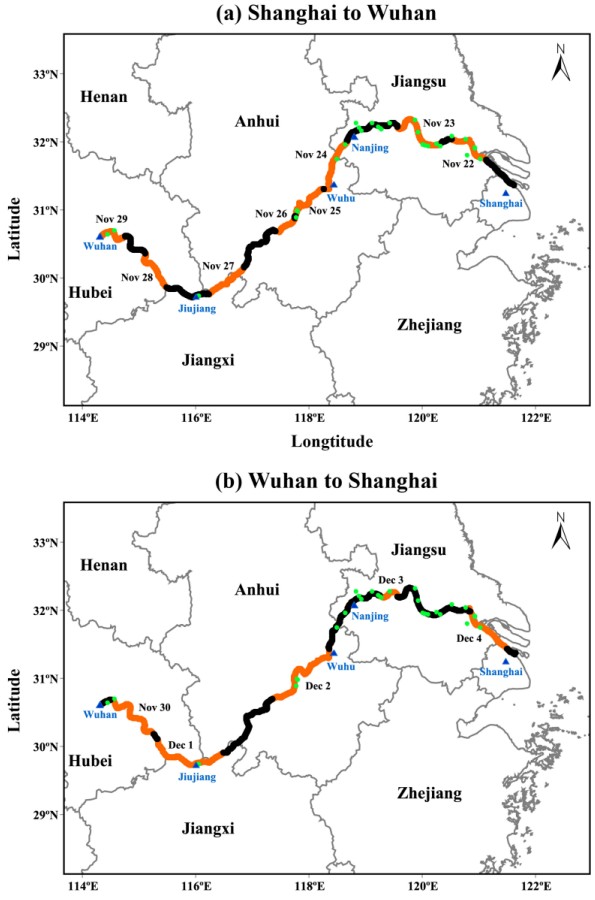

**Figure 1.** The ship tracks of (a) the departing journey (from Shanghai to Wuhan) and (b) the returning journey (from Wuhan to Shanghai). The sections of the cruise track highlighted in orange indicate the period of MAX-DOAS observations. The green dots represent the major power plants along the Yangtze River.




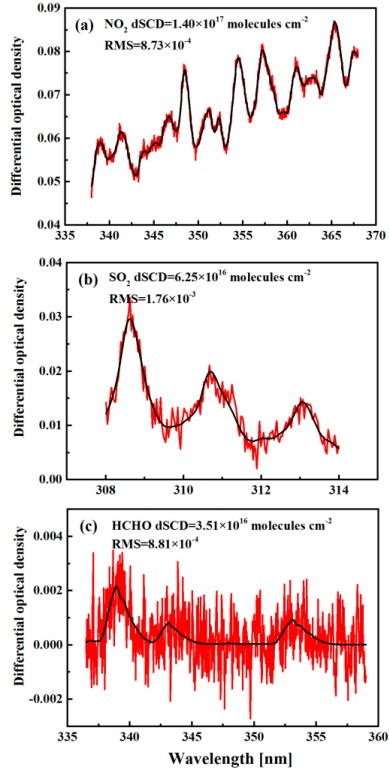

**Figure 2.** An example of DOAS fit for (a) NO$_2$, (b) SO$_2$, and (c) HCHO. The spectrum was taken on 1 December 2015 at
14:02 LT with elevation angle of 30°. Red lines show the measured atmospheric spectrum after all other absorbers have been
subtracted, and the black line shows the scaled reference absorption cross section.





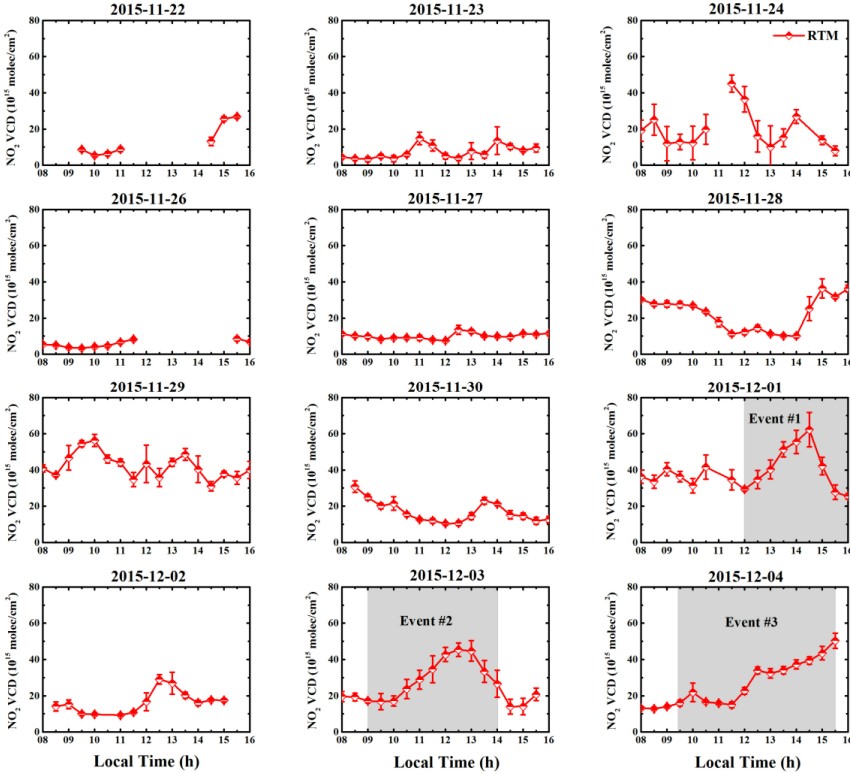

**Figure 3.** Time series of NO$_2$ vertical column densities (VCDs) during Yangtze River campaign. The error bars refer to the

1σ variation of the measurement. The gray area highlighted the episode periods.

Notes: The data were half-hour averages and individual measurements taken under unfavorable wind directions have been

filtered before averaging. No data presented on 25 November due to frequent power outages.



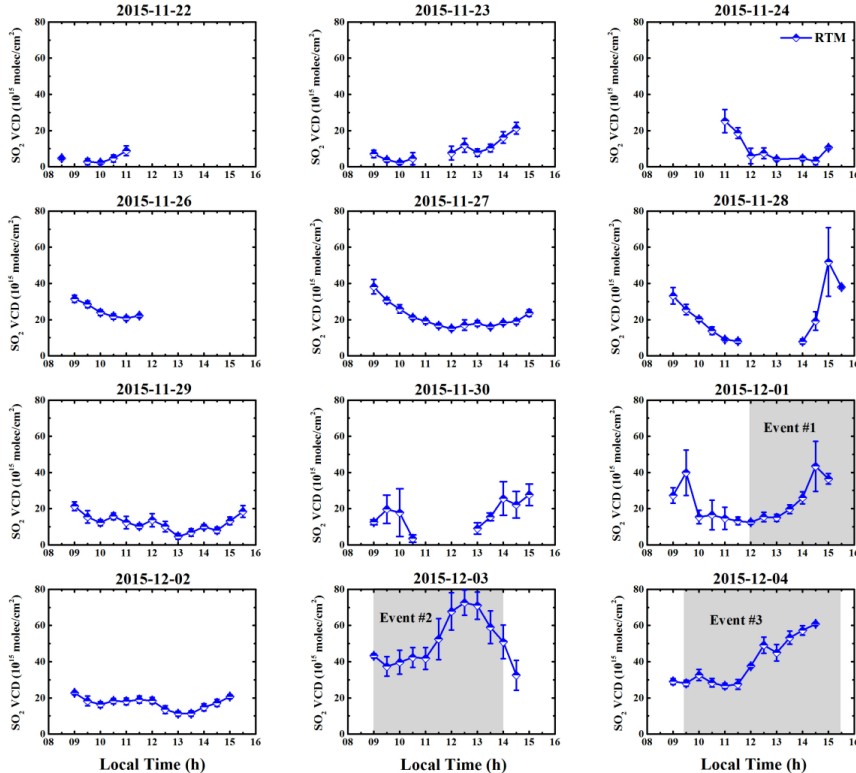

**Figure 4.** Same as Fig. 3, for the time series of SO$_2$ vertical column densities (VCDs).



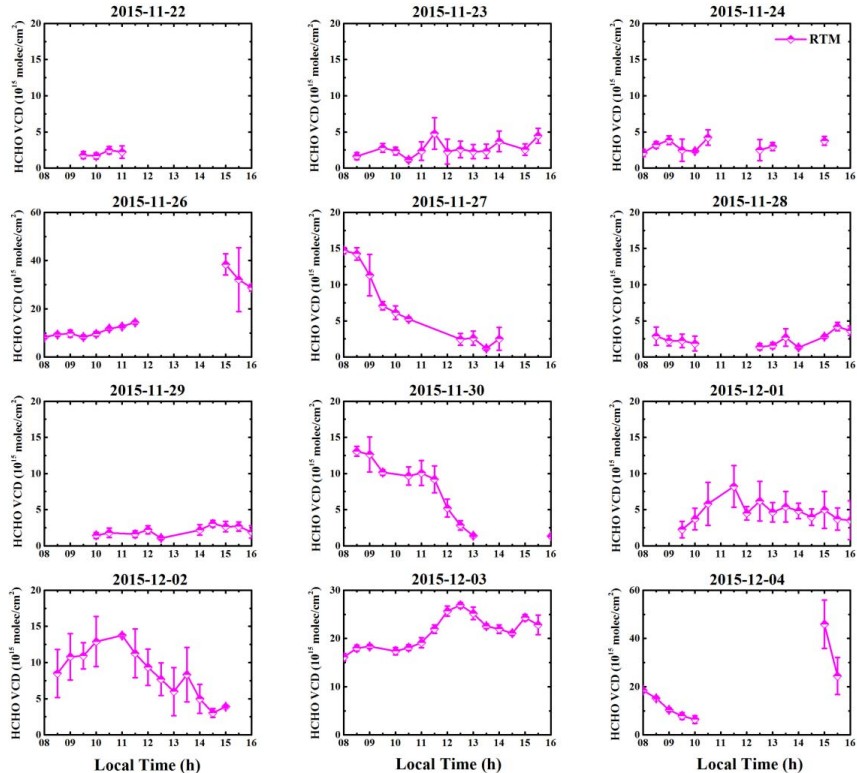

**Figure 5.** Same as Fig. 3, but for the time series of HCHO vertical column densities (VCDs). The y-axis scale of each panel is different.






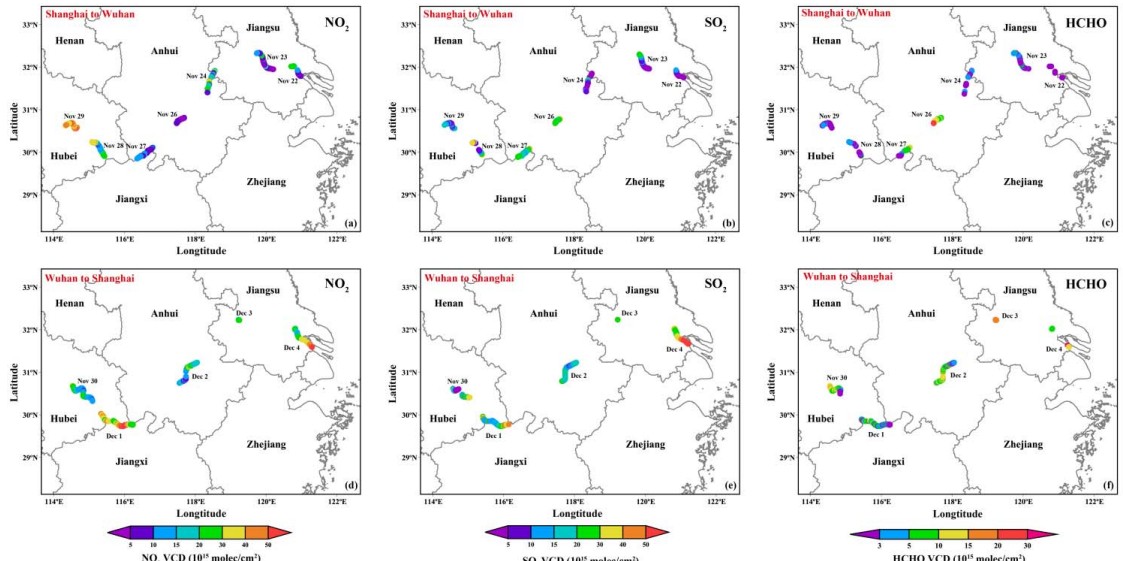

**Figure 6.** Spatial distribution of tropospheric $NO_2$ (left panels), $SO_2$ (middle panels), and HCHO (right panels) VCDs along the departing route (upper panels, a-c, from Shanghai to Wuhan) and returning route (lower panels, d-f, from Wuhan to Shanghai).

Notes: The ship was anchored at Yizheng Marine department on 3 December 2015.





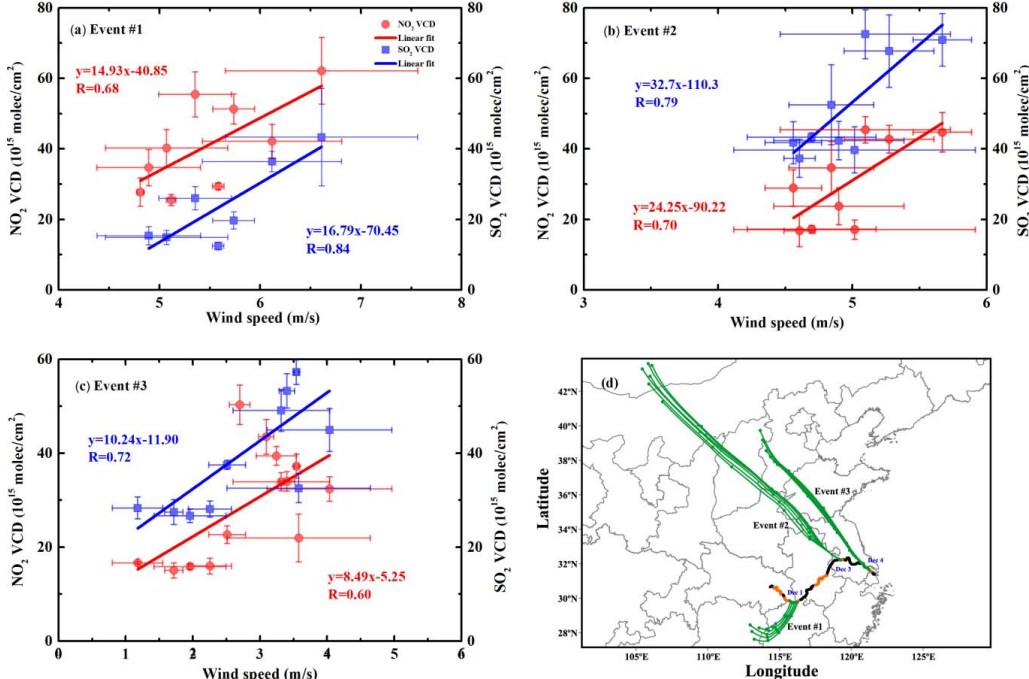

**Figure 7.** (a-c) Correlation analysis of ship-based MAX-DOAS VCDs (red circle show the NO$_2$ VCDs and blue square show the SO$_2$ VCDs) and wind speed for three high VCDs events. (d) Yangtze River Cruise Track and 24 h backward trajectories

calculated by the NOAA HYSPLIT model for three events (green marks indicate starting point, −6, −12, −18 and −24 h, respectively). The data were half-hour averages. The error bars show the 1σ standard deviations of mobile DOAS VCDs and wind speed.




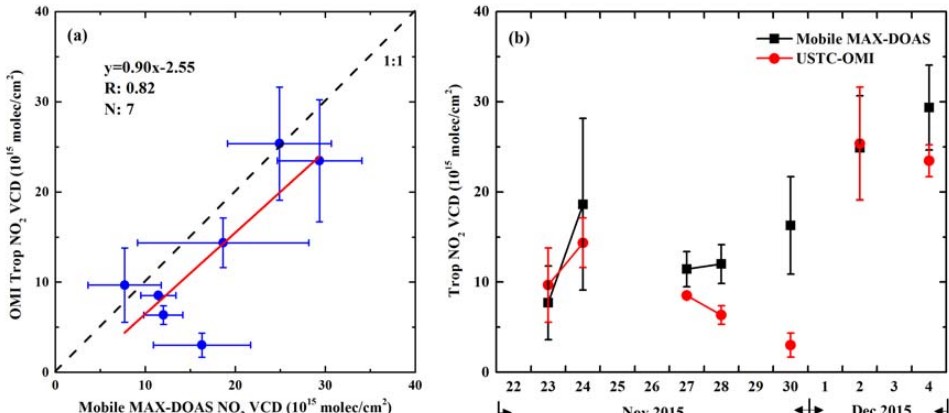

**Figure 8.** (a) Correlation analysis and (b) time series of tropospheric $NO_2$ VCDs measured by ship-based MAX-DOAS and
OMI during Yangtze River campaign. MAX-DOAS data (black markers) are temporally averaged around the OMI overpass
time, while the OMI data (red markers) are spatially averaged within 20 km radius around the ship's averaged position. The
error bars show the $1\sigma$ standard deviations of ship-based MAX-DOAS and OMI data.



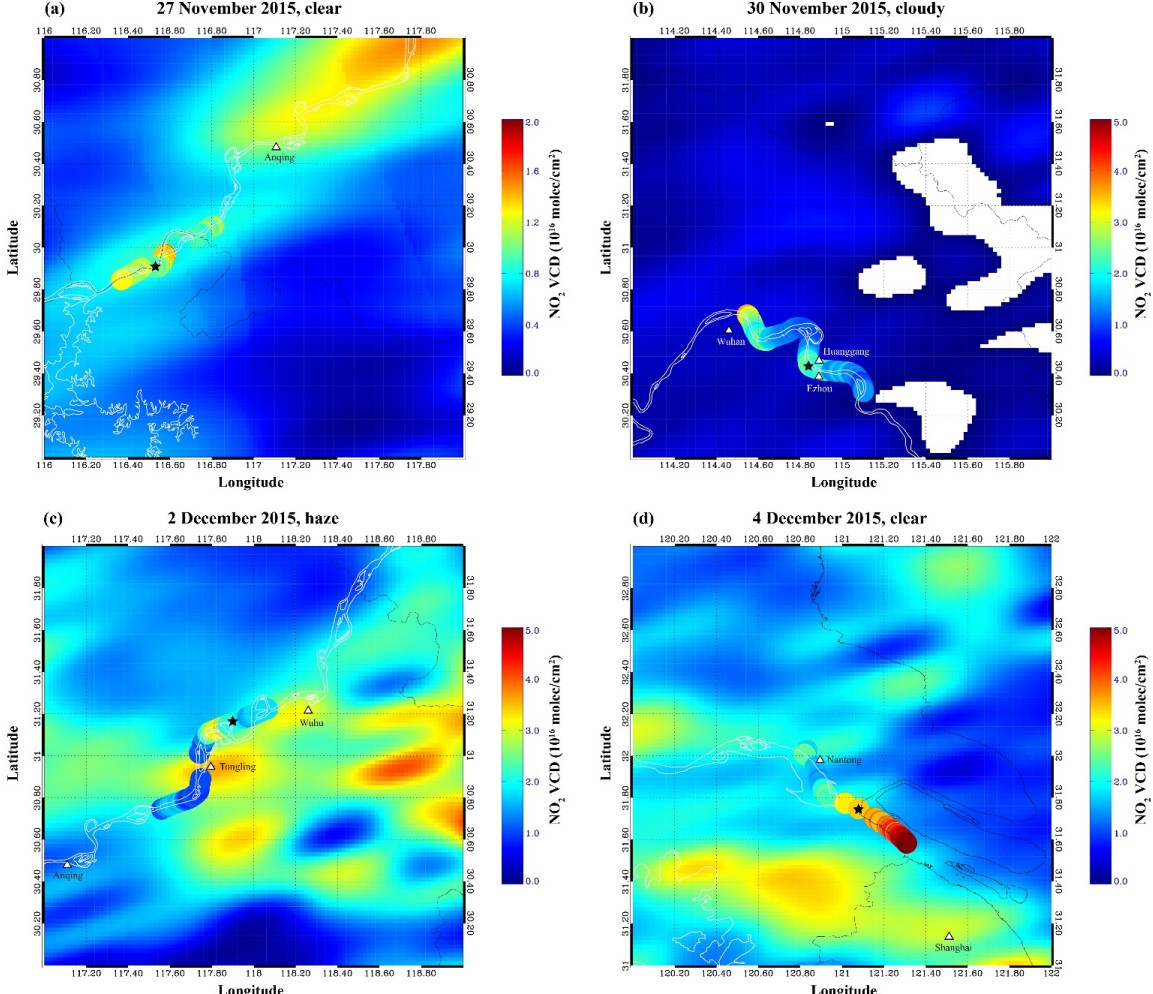


**Figure 9.** Spatial pattern of tropospheric NO$_2$ VCD measured by ship-based MAX-DOAS and OMI during four typical days within the Yangtze River campaign. The color-coded circle indicates the ship-based MAX-DOAS observations. Each plot show example for relatively clear (a and d), cloudy (b), and haze (c) metrological conditions along Yangtze River. The star symbols indicate the ship position during the OMI overpass time (~13:45 LT). The colorbar scale of each panel is different.


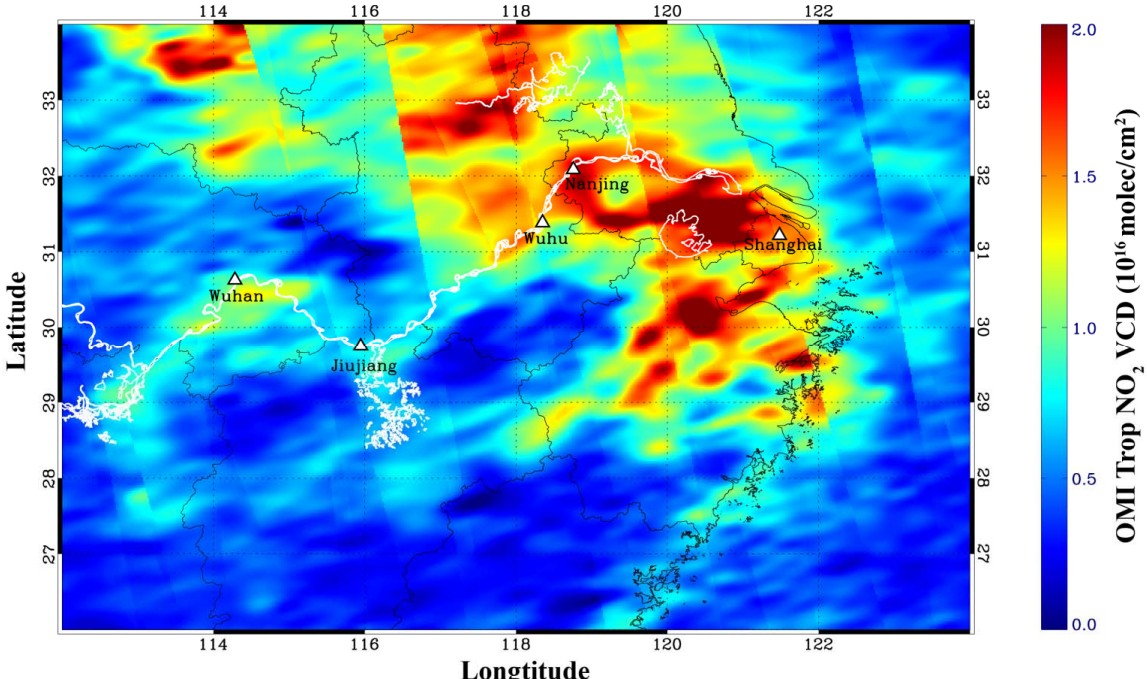

**Figure 10.** The spatial distribution of averaged tropospheric NO₂ VCDs measured by OMI during Yangtze River campaign (22 November to 4 December, 2015).




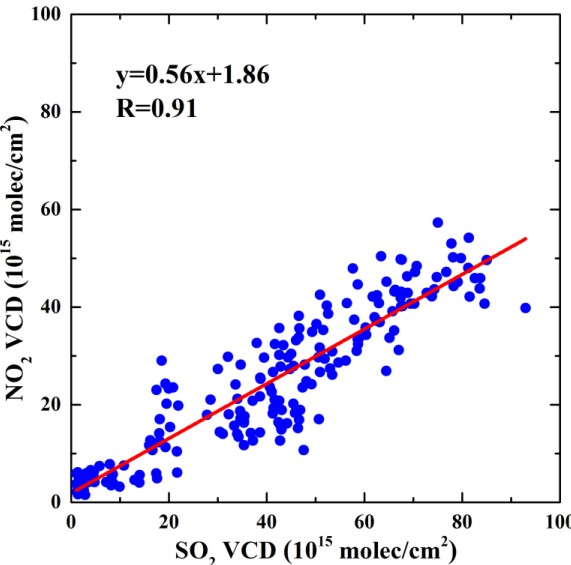


**Figure 11.** Scatter plot of NO$_2$ and SO$_2$ VCDs data around power plants along the Yangtze River. NO$_2$ and SO$_2$ VCDs measured within 2 km of the power plants are used for the analysis.



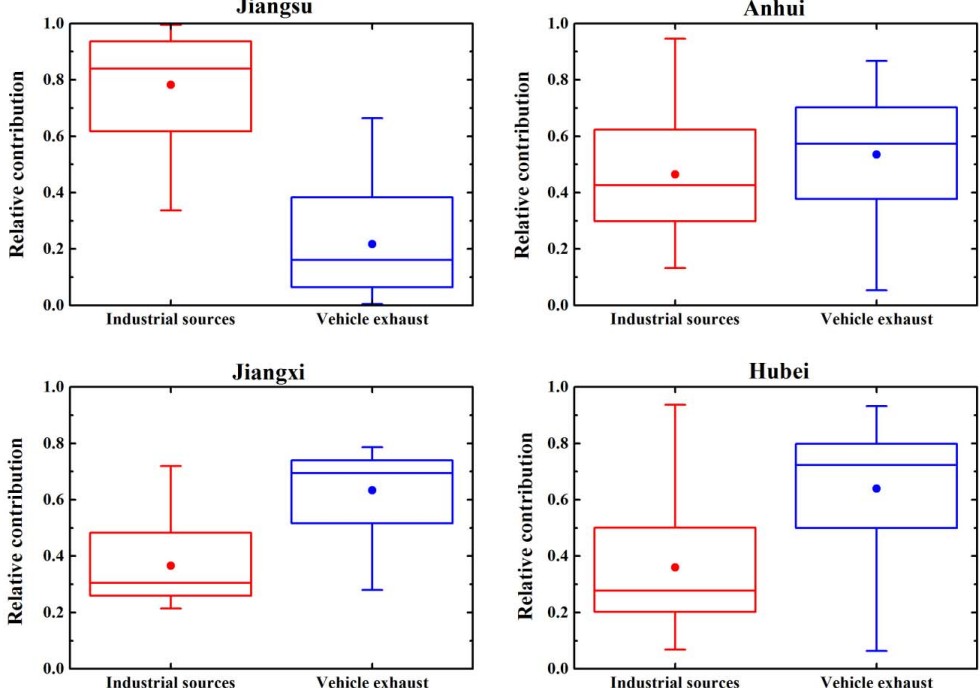

**Figure 12.** Relative contributions of industrial sources and vehicle exhaust to the ambient NO$_2$ levels over the four provinces.

Notes: the bottom and top of the box represent the 25th and 75th percentiles, respectively; the line within the box represent

the median; the dot represents the mean; the whiskers below and above the box stands for the 10th and 90th percentiles.





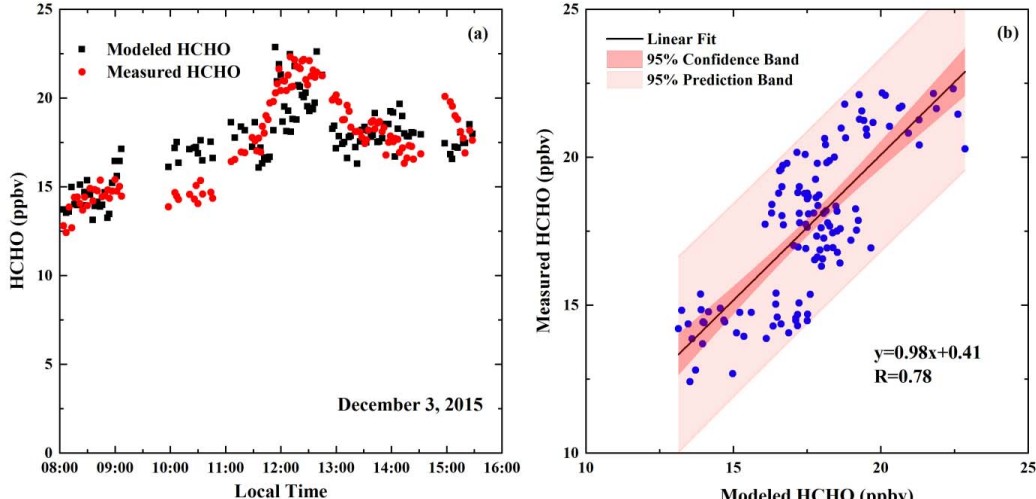

**Figure 13.** Comparison of the measured and modeled HCHO values from multiple linear regression on 3 December 2015.
The left panel shows the modeled and measured HCHO time series. The right panel shows the linear correlation between the
modeled and measured HCHO concentrations. The black solid line indicates the linear regression. The red and pink areas
denote the 95% confidence interval and the 95% prediction, respectively.





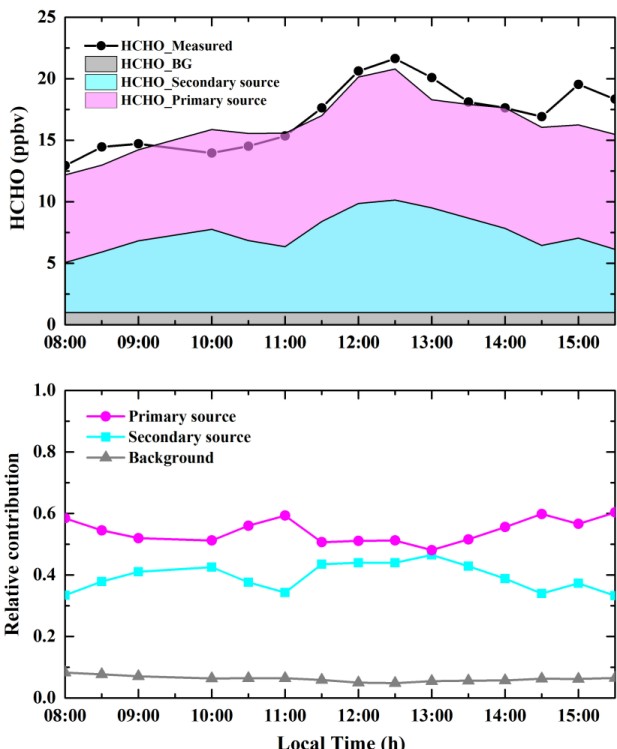


**Figure 14.** Time series of absolute (upper) and relative contribution (lower) of primary source, secondary source, and atmospheric background to ambient HCHO level on 3 December 2015.