# Peer review of "Ship-based MAX-DOAS measurements of tropospheric NO2, SO2, and HCHO distribution along the Yangtze River"

_Atmospheric Chemistry and Physics, 2017_

## Referee Comment (RC1) · Anonymous Referee #1 · 28 Jan 2018

The manuscript presents ship-based MAX-DOAS measurements along the Yangtze River to investigate the tropospheric pollutants distribution and their sources over eastern China. The study combines MAX-DOAS, lidar, satellite as well as in-situ sensors measurements to assess the spatial distribution and source appointment of pollutants along Yangtze River during winter time of 2015. The MAX-DOAS VCDs obtained were used to valid OMI satellite observations. The authors also use the NO2 to SO2 ratio to estimate the industrial and traffic emission over different areas along the river. In addition, primary and secondary sources of HCHO, an indicator of VOCs, were estimated using a simple regression model. In general, the topic is interesting and fitting well to the scope of journal. The manuscript is well written with clear scientific objectives. I

recommend publication after addressing the comments provided below.

Comments:

1) The authors use lidar aerosol profile for AMF calculations to convert SCDs to VCDs for trace gases. However, only aerosol profile at the lowest 2 km is used and aerosol above 2 km are ignored. I think it is necessary to estimate the uncertainty due to ignoring aerosols above 2 km. In addition, the lidar has a blind height of 195m, so how the aerosol at the lowest 195m were treated?

2) As described in Sect 2.2.4, the authors has finally employed the new method for VCD estimation in the mobile measurements, which is recommended by Wagner et al., 2010. Maybe the authors could shorten the introduction of the geometric approximation and standard method for the VCD estimation. Alternatively, I suggest the authors to provide the comparison of the retrieved VCD results between standard and suggested methods? For example, taking one day as an example, to present the time series of the DSCD_mea, DSCD_offset and AMF_trop as used in the E.q. (12).

3) In this study, OMI VCDs are computed using atmospheric profiles from WRF-Chem simulations. It would be interesting to show how the VCDs are different from other operational products, e.g., NASA product.

4) It is not clear how the pollution events are identified. From Figure 3, the NO2 VCD on 29 of Nov is also very high, however, this day is not identified as pollution event. The authors should provide more information on the criterion in selecting pollution events.

5) In section 3.1, trajectories are calculated to assess the pollutant transportation. However, it is not clear that these backward trajectories are calculated at which altitude level. It would be much better to show the height of the backward trajectories as well, so that readers could better intemperate how pollutants are transported.

6) It's novel and interested to identify the industrial and vehicle contribution by E.q. 13 and 14, which provide the new insights for the sources appointment. As one of

Interactive
comment
the most developed area in China, there were some more studies focusing on the emission inventories and source appointment for YRD areas. Maybe the authors can review previous results and compare with this study a little bit more in the discussion.

7) For the estimation of primary and secondary sources of HCHO, I have two questions: 1) which kind of the measured HCHO concentrations was used in the regression model? Since the HCHO levels were determined by MAX-DOAS as VCDs, however, the ambient HCHO concentrations were used in the model. How the authors obtain the ground surface concentration HCHO from the VCDs? Otherwise, why the authors can make regression of the HCHO VCDs with in-situ CO, Ox? 2) For the diurnal pattern, the authors inferred that secondary formation of HCHO shows a peak value during noon time (11:00-14:00), however, there was also another peak of relative contribution of secondary sources around 10:00 LT. How to consider this phenomenon?

Technical corrections:

1) Introduce abbreviation on the first time used in the manuscript, e.g., line 169 WRF-Chem.

2) Text on Figure 3-6 is too small to read. Considering the similarities of Fig.3 to Fig.5, I suggest to merge them together to show the time series of these three pollutants in each panel with a continuous X-axis.

3) Put error bars on Figure 13 and 14

4) Please uniform the units of VCDs.   It's different in Figures (molec/cm2) and manuscript text (molec cm-2) now.

---

## Referee Comment (RC2) · Anonymous Referee #2 · 19 Mar 2018

The manuscript investigates tropospheric NO2, SO2, and HCHO characteristic along the Yangtze River in winter 2015 using ship-based MAX-DOAS measurements. It provides a better insight of tropospheric pollutants distribution and their sources over eastern China. The MAX-DOAS NO2 VCDs were compared to OMI satellite observations. In addition, NO2 to SO2 ratio was used to estimate the relative contributions of industrial sources and vehicle emissions to ambient NO2 levels. Finally, the authors use a multiple linear regression model to estimate the contribution of primary and secondary sources of HCHO. In general, the manuscript is well written and organized. I think the topic and the findings fit well to the journal scope. However, there are still some issues need to be addressed before publication. My comments are listed below.

[Figure]

Comments:

1. In section 2.2.2, how many percent of data are removed using the filter wind directions and SZAs?

2. The authors use the trace gas profiles and vertical profiles of pressure and temperature from WRF-Chem for AMF calculation. What is the spatial resolution and time resolution of the WRF-Chem profiles? Did the authors use a fixed trace gases profile for all the measurements or use the unique profile dependent on the measurement locations and time?

3. The authors use the aerosol extinction coefficients measured by Mie lidar measurements for the AMF calculations. Did the authors use a fixed aerosol extinction profile (average the aerosol extinction coefficient from all the lidar measurements during the campaign) or use the specified profile dependent on locations, time and the availability of lidar measurements?

4. A fix set of single scattering albedo (SSA) of 0.95, asymmetry parameter of 0.68 and surface albedo of 0.06 is assumed in the radiative transfer calculations. Please explain why use this setting. Any references?

5. In section 2.2.4, there are too many formula and introduction of the determination of the tropospheric VCD. I suggest shortening the section and combining some of the formula.

6. I agree with the comment from Referee #1 about the high NO2 VCD on Nov. 29. Please explain the reason of high NO2 and relative low SO2 concentrations during this day.

7. In section 2.3, USTC's OMI tropospheric NO2 product is used. I suggest showing more detailed information about the USTC OMI product, e.g., the data source of NO2 slant density (SCD). In addition, USTC shall be explained when appeared for the first time.

8. Figure 7d is a bit confusing, please describe clearly what the green lines represent? Is it 6, 12, 18 or 24 hour backsward trajectories? What is the altitude of the backward trajectories?

9. The authors mention that the different spatial patterns detected by MAX-DOAS and OMI on Dec. 2 might be due to the strong influence by the aerosols. This is an interesting episode because it might show that the effect of neglecting aerosol in satellite AMF calculations on satellite NO2 retrieval. Please prove this hypothesis using the Lidar measurement data.

10. In section 3.1, Line 307-308: "In contrast, lower HCHO VCDs were observed mainly on rainy, cloudy and haze days". Could the authors explain more about the possible reasons for this phenomenon?

11. In section 3.4, please explain how to convert HCHO VCD to HCHO concentrations (ppb).

12. In section 3.4, Line 430: "As other factors can also affect the atmospheric HCHO concentration…". Please describe the "other factors" in more detail.

Technical corrections:

There are still some typos in the manuscript. In addition, the use of the notations should also be checked carefully. A few examples of these technical errors are listed below:

1. Line 321: 'Dec 1 and 3' > 'Dec 2 and 4'

2. Line 344: 'aerosol' > 'aerosols'

3. Text on Figure 2: change "dSCD" to "DSCD", and please uniform the units of DSCD (molec/cm2).

4. Explain the caption 'RTM' in Figure 3-5. Or remove it.

5. Text on Figure 6 is too small to read. Please use larger font size and put the colorbar

on the right.

6. Improve the quality of Figure 10 (too many stripes)

---

## Author Comment (AC1) · 11 Apr 2018

**Responses to Anonymous Referee #1**

We thank the reviewer for the constructive suggestions and comments. We appreciate the reviewer's comments and these comments are very helpful for improving the manuscript. We understand that the comments are positive on the scientific content of the manuscript while appropriate revisions and clarifications are necessary.

We have addressed the reviewer's comments on a point to point basis as below for consideration.

**General comments:**

1) The authors use lidar aerosol profile for AMF calculations to convert SCDs to VCDs for trace gases. However, only aerosol profile at the lowest 2 km is used and aerosol above 2 km are ignored. I think it is necessary to estimate the uncertainty due to ignoring aerosols above 2 km. In addition, the lidar has a blind height of 195m, so how the aerosols at the lowest 195m were treated?

**Response:** A sensitivity study has been conducted to address the influence from aerosol above 2km to the AMF calculation. Tropospheric NO2, SO2 and HCHO AMFs are calculated using lidar aerosol profile at the lowest 2km and WRF-Chem aerosol profile above 2km. The results are compared to AMFs calculated with only lidar aerosol profile at the lowest 2km. Comparison results show that AMFs calculated with considering aerosol above 2km are on average 2-4% lower than AMFs without considering aerosol at upper altitudes. The result indicates that ignoring aerosols above 2 km only cause a negligible error on the AMF calculation. In addition, aerosols at the lowest 195m are considered to be homogeneous in the AMF calculation. These information are now supplemented in the manuscript line 174 to 180.

2) As described in Sect 2.2.4, the authors has finally employed the new method for VCD estimation in the mobile measurements, which is recommended by Wagner et al., 2010. Maybe the authors could shorten the introduction of the geometric approximation and standard method for the VCD estimation. Alternatively, I suggest the authors to provide the comparison of the retrieved VCD results between standard and suggested methods? For example, taking one day as an example, to present the time series of the DSCD\_mea, DSCD\_offset and AMF\_trop as used in the E.q. (12).

**Response:** We followed the reviewer suggestion and shortened the introduction of the geometric approximation and standard method for the VCD estimation. Changes are listed in the following:

- a) Line 149-150: we put the formula ( $SCD_{meas} = SCD_{trop} + SCD_{strat}$ ) into the sentence
- b) Line 151-152: sentence combination: "it can be assumed that the light path in the stratosphere for zenith and off zenith measurements are very similar, i.e. SCDstrat(α)≈SCDstrat(90°)"
- c) Line 157-158: add description of VCDtrop: "and the tropospheric vertical column density (VCDtrop) can be expressed as follows"

d) Line 193-195: remove this paragraph in the ACPD version

In addition, we have compared the VCD retrieved with both standard and suggested method. Time series of the DSCD\_meas, DSCD\_offset, AMF\_trop and VCD\_trop on 28 Nov 2015 are shown in Figure R1. Fig. R1a shows the time series of NO2 DSCDs for both elevation angles. Higher DSCDs are obtained from the lower elevation angle. The offset caused by the NO2 absorption in the Fraunhofer reference spectrum and the stratospheric absorption (see Eq. 7 in the revised manuscript) are determined by a polynomial fit. DSCDoffset(t) is then calculated following Eq. 10. The tropospheric VCDs of NO2 calculated with both approaches (corresponding to Eq. 7 and Eq. 8 in the revised manuscript) are shown in Fig. R1d. Comparing with standard method, the temporal development of tropospheric NO2 VCDs calculated by the new method is less noisy. Similar results can be found for SO2 and HCHO in Figs. R2-3.

Figure R1. An example of determination of tropospheric  $NO_2$  from the spectra measured on 28

November 2015. (a) NO2 DSCDs, (b) DSCDoffset (see Eq. 10) plotted as a function of time (green points), (c) Tropospheric NO2 AMFs calculated by the radiative transfer model SCIATRAN, (d) The tropospheric VCDs of NO2 calculated by new and standard method (corresponding to Eq. 7 and Eq. 8).

---

## Author Comment (AC2) · 11 Apr 2018

**Responses to Anonymous Referee #2**

We thank the reviewer for the constructive suggestions and comments. We appreciate the reviewer's comments and these comments are very helpful for improving the manuscript. We understand that the comments are positive on the scientific content of the manuscript while appropriate revisions and clarifications are necessary.

We have addressed the reviewer's comments on a point to point basis as below for consideration.

**General comments:**

1) In section 2.2.2, how many percent of data are removed using the filter wind directions and SZAs?

**Response:** The wind direction and SZA filtering criterions removed 5.4% and 15.8% of all data, respectively. This information is now supplemented in the revised manuscript line 122 to 123.

2) The authors use the trace gas profiles and vertical profiles of pressure and temperature from WRF-Chem for AMF calculation. What is the spatial resolution and time resolution of the WRF-Chem profiles? Did the authors use a fixed trace gases profile for all the measurements or use the unique profile dependent on the measurement locations and time?

**Response:** In this study, the WRF-Chem model (version 3.7) was used to simulate the vertical profile of aerosol and trace gases as well as other meteorological parameters. The simulation domain covers large part of East China (17-49 °N, 95-124°E) with a horizontal resolution of  $20 \times 20$  km and 26 pressure sigma level from ground level up to50 hPa. The time resolution of the model output is set to 1 h. Details of the configuration of the model can be found in (Liu et al., 2016;Su et al., 2017). Atmospheric profiles obtained from the model simulation were then interpolated in both spatial and temporal dimension to MAX-DOAS measurements location and time for AMF calculation. Details of the model simulation and AMF calculation are now included in the manuscript line 171 to 172.

3) The authors use the aerosol extinction coefficients measured by Mie lidar measurements for the AMF calculations. Did the authors use a fixed aerosol extinction profile (average the aerosol extinction coefficient from all the lidar measurements during the campaign) or use the specified

profile dependent on locations, time and the availability of lidar measurements?

**Response:** Lidar measurement was carried out together with MAX-DOAS most of the time during the campaign. AMF of the MAX-DOAS observations is calculated using hourly averaged Lidar aerosol extinction profile. This information is now supplemented in the manuscript line 172.

4) A fix set of single scattering albedo (SSA) of 0.95, asymmetry parameter of 0.68 and surface albedo of 0.06 is assumed in the radiative transfer calculations. Please explain why use this setting. Any references?

**Response:** The single scattering albedo (SSA) of 0.95 and asymmetry parameter of 0.68 are chosen according to the sensitivity study by (Chen et al., 2009). For oceans or rivers, the surface albedo is generally low and keep around 0.06 (Pinker et al., 1995). We have added the references in the revised manuscript line 188.

5) In section 2.2.4, there are too many formula and introduction of the determination of the tropospheric VCD. I suggest shortening the section and combining some of the formula.

**Response:** We followed the reviewer suggestion and shortened the introduction of the determination of the tropospheric VCD and combined some of the formula. Changes are listed in the following:

a) Line 149-150: we put the formula (SCDmeas = SCDtrop + SCDstrat) into the sentence

b) Line 151-152: sentence combination: "it can be assumed that the light path in the stratosphere for zenith and off zenith measurements are very similar, i.e.  $SCDstrat(\alpha) \approx SCDstrat(90^{\circ})$ "

c) Line 157-158: add description of VCDtrop: "and the tropospheric vertical column density (VCDtrop) can be expressed as follows"

d) Line 193-195: remove this paragraph about the description of tropospheric vertical column density (VCDtrop) in the ACPD version

e) Line 207: delete the "tropospheric vertical column density"

6) I agree with the comment from Referee #1 about the high NO2 VCD on Nov. 29. Please explain the reason of high NO2 and relative low SO2 concentrations during this day.

**Responses:** The measurement ship was sailing around Wuhan on 29 November. Elevated tropospheric  $NO_2$  observed around Wuhan is probably related to high emission, i.e., traffic emissions, in Wuhan as it is the largest city in Hubei. A rather low  $SO_2$  level might due to lower  $SO_2$  emissions located along the Yangtze River around Wuhan, e.g., coal-fired power plants.

7) In section 2.3, USTC's OMI tropospheric  $NO_2$  product is used. I suggest showing more detailed information about the USTC OMI product, e.g., the data source of  $NO_2$  slant density (SCD). In addition, USTC shall be explained when appeared for the first time.

**Response:** We agree with the reviewer suggestion and now included a more detailed description of the USTC's OMI NO2 product in section 2.3. In this study, USTC's OMI tropospheric NO2 product is developed based on OMI's primary product and has proven to be more suitable for the atmospheric conditions in China (Liu et al., 2016;Su et al., 2017;Xing et al., 2017). Slant column densities (SCDs) of NO2 are retrieved by applying the DOAS fit to OMI spectra (data source: OMI Level 1**B** VIS Global Radiances Data product (OML1BRVG) (https://disc.gsfc.nasa.gov/Aura/data-holdings/OMI/oml1brvg\_v003.shtml)). The information of USTC's OMI product are now supplemented in the revised manuscript line 234-237.

8) Figure 7d is a bit confusing, please describe clearly what the green lines represent? Is it 6, 12, 18 or 24 hour backward trajectories? What is the altitude of the backward trajectories?

**Response:** In Figure 7d, the green lines represent the 24 h backward trajectories during pollution events. The green lines show the backward trajectories calculated for each hour during the detected episodes. For example, five green lines on 1 Dec represent the backward trajectories ending at 04, 05, 06, 07, and 08 UTC on 1 Dec 2015 (corresponding to gray columns in Fig 4). The green markers indicate the location of air masses 6, 12, 18 and 24 h before arriving to the measurement ship. Considering atmospheric pollutants are mainly concentrated in low altitudes during heavy pollution episodes, the trajectory arrival heights were set to 500 m and assumed to be representative for the entire boundary layer. This information is now included in the manuscript line 300-302.

9) The authors mention that the different spatial patterns detected by MAX-DOAS and OMI on

Dec. 2 might be due to the strong influence by the aerosols. This is an interesting episode because it might show that the effect of neglecting aerosol in satellite AMF calculations on satellite  $NO_2$ retrieval. Please prove this hypothesis using the Lidar measurement data.

**Response:** We agree with the reviewer suggestion and estimated the influence of ignoring aerosols in satellite AMF calculation on NO2 retrieval. As lidar measurements only cover limited area, therefore, we have recalculated OMI NO2 VCDs by taking aerosol information from WRF-Chem simulation into account. Figure R1a shows OMI NO2 VCDs calculated without consider aerosol in the AMF calculation while Figure R1b shows the OMI NO2 VCDs recalculated using aerosol information from WRF-Chem simulation. The spatial distributions of NO2 are changed after including aerosol in the radiative transfer calculation. Significant enhancement of NO2 VCDs can be observed over some areas. However, the spatial patterns of NO2 detected by MAX-DOAS and OMI are still quite different. The result indicates the impact of aerosol could not fully explain the discrepancy between MAX-DOAS and OMI on this day. On the other hand, the MAX-DOAS and OMI observations agree better during OMI overpass time (black star symbols) and the agreement decay when the time differences between MAX-DOAS and OMI measurements become larger. This implies a strong temporal variability of NO2 on this day. We have now supplemented the additional information and explanation in the manuscript line 352-360.

**Figure R1.** Spatial pattern of tropospheric NO2 VCD measured by OMI calculated under two different AMFs.

10) In section 3.1, Line 307-308: "In contrast, lower HCHO VCDs were observed mainly on rainy,

cloudy and haze days". Could the authors explain more about the possible reasons for this phenomenon?

**Response:** Lower HCHO VCDs were observed mainly on rainy, cloudy and haze days, which might be due to stronger wet deposition and weaker solar irradiation during these days. The possible reasons for this phenomenon are supplemented in the revised manuscript line 316-318.

11) In section 3.4, please explain how to convert HCHO VCD to HCHO concentrations (ppb).

**Response:** For the estimation of primary and secondary sources of HCHO, the measured HCHO used in the regression model is ground mixing ratios. Usually, surface HCHO mixing ratios can be obtained from the HCHO vertical profiles (e.g., 0-200 m layer) (Wang et al., 2014). As the viewing elevation angles of the MAX-DOAS measurements only include 30° and 90°, therefore, there is not enough information to retrieve HCHO vertical profiles. In this study, we use a simplified formula introduced by (Lee et al., 2008) to convert mean HCHO DSCDs to mixing ratios (ppbv).

$$M(\text{ppbv}) = 1.25 \times \frac{DSCD(\text{molecule } \text{cm}^{-2})}{dAMF} \times \frac{1}{2.688 \times 10^{16} (\text{molecule } DU^{-1})} \times \frac{1}{\Delta P(\text{atom})}$$

where M is the mixing ratio, DSCD is the difference between the SCDs of the measured spectrum and that of the Fraunhofer reference spectrum, dAMF is a differential air mass factor (dAMF=AMF( $\alpha$ =30°)-AMF( $\alpha$ =90°)), and  $\Delta$ P is the pressure difference between surface and 500m height of boundary layer. The AMFs for this study were calculated using the radiative transfer model SCIATRAN 2.2 as described in Section 2.2.4. These information are now supplemented in the revised manuscript line 433-434.

12) In section 3.4, Line 430: "As other factors can also affect the atmospheric HCHO concentration". Please describe the "other factors" in more detail.

**Response:** In addition to primary emission and secondary formation of HCHO, meteorological conditions, e.g., solar irradiance, could also affect the atmospheric HCHO concentration. This information is now included in the revised manuscript line 451.

**Technical corrections:**

1) Line 321: 'Dec 1 and 3'> 'Dec 2 and 4'

Response: Corrected.

2) Line 344: 'aerosol' > 'aerosols'

Response: Corrected.

3) Text on Figure 2: change "dSCD" to "DSCD", and please uniform the units of DSCD (molec/cm2).

**Response:** We have changed "dSCD" to "DSCD" on Figure 2 and uniform the units of DSCD (molec/cm2) in the revised manuscript.

4) Explain the caption 'RTM' in Figure 3-5. Or remove it.

**Response:** We have removed the caption 'RTM' in Figures 3-5. We have merged Figs.3-5 so that the time series of three pollutants (NO2, SO2, and HCHO) in each panel with a continuous X-axis (Fig. 3&4 in the revised manuscript).

5) Text on Figure 6 is too small to read. Please use larger font size and put the colorbar on the right.

**Response:** We have enlarged the fonts in Figure 6 and put the colorbar on the right (Fig. 5 in the revised manuscript).

6) Improve the quality of Figure 10 (too many stripes).

**Response:** We have do our best to improve the quality of Figure 10, but too many stripes in Fig. 10 is inevitable which caused by the different satellite orbit.

**References:**

- Chen, D., Zhou, B., Beirle, S., Chen, L. M., and Wagner, T.: Tropospheric NO2 column densities deduced from zenith-sky DOAS measurements in Shanghai, China, and their application to satellite validation, Atmospheric Chemistry And Physics, 9, 3641-3662, 2009.
- Lee, C., Richter, A., Lee, H., Kim, Y. J., Burrows, J. P., Lee, Y. G., and Choi, B. C.: Impact of transport of sulfur dioxide from the Asian continent on the air quality over Korea during May 2005, Atmospheric Environment, 42, 1461-1475, 2008.
- Liu, H., Liu, C., Xie, Z., Li, Y., Huang, X., Wang, S., Xu, J., and Xie, P.: A paradox for air pollution controlling in China revealed by "APEC Blue" and "Parade Blue", Scientific reports, 6, doi:10.1038/srep34408, 2016.
- Pinker, R., Frouin, R., and Li, Z.: A review of satellite methods to derive surface shortwave irradiance, Remote Sensing of Environment, 51, 108-124, 1995.
- Su, W., Liu, C., Hu, Q., Fan, G., Xie, Z., Huang, X., Zhang, T., Chen, Z., Dong, Y., and Ji, X.: Characterization of ozone in the lower troposphere during the 2016 G20 conference in Hangzhou, Scientific reports, 7, 17368, 2017.
- Wang, T., Hendrick, F., Wang, P., Tang, G., Clémer, K., Yu, H., Fayt, C., Hermans, C., Gielen, C., and Müller, J.-F.: Evaluation of tropospheric SO2 retrieved from MAX-DOAS measurements in Xianghe, China, Atmospheric Chemistry and Physics, 14, 11149-11164, 2014.
- Xing, C., Liu, C., Wang, S., Chan, K. L., Gao, Y., Huang, X., Su, W., Zhang, C., Dong, Y., and Fan, G.: Observations of the vertical distributions of summertime atmospheric pollutants and the corresponding ozone production in Shanghai, China, Atmospheric Chemistry and Physics, 17, 14275-14289, 2017.

---

## Author Response (AR1)

[revised manuscript text omitted]

**Responses to Anonymous Referee #1**

We thank the reviewer for the constructive suggestions and comments. We appreciate the reviewer's comments and these comments are very helpful for improving the manuscript. We understand that the comments are positive on the scientific content of the manuscript while appropriate revisions and clarifications are necessary.

We have addressed the reviewer's comments on a point to point basis as below for consideration.

**General comments:**

1) The authors use lidar aerosol profile for AMF calculations to convert SCDs to VCDs for trace gases. However, only aerosol profile at the lowest 2 km is used and aerosol above 2 km are ignored. I think it is necessary to estimate the uncertainty due to ignoring aerosols above 2 km. In addition, the lidar has a blind height of 195m, so how the aerosols at the lowest 195m were treated?

**Response:** A sensitivity study has been conducted to address the influence from aerosol above 2km to the AMF calculation. Tropospheric $NO_2$, $SO_2$ and HCHO AMFs are calculated using lidar aerosol profile at the lowest 2km and WRF-Chem aerosol profile above 2km. The results are compared to AMFs calculated with only lidar aerosol profile at the lowest 2km. Comparison results show that AMFs calculated with considering aerosol above 2km are on average 2-4% lower than AMFs without considering aerosol at upper altitudes. The result indicates that ignoring aerosols above 2 km only cause a negligible error on the AMF calculation. In addition, aerosols at the lowest 195m are considered to be homogeneous in the AMF calculation. These information are now supplemented in the manuscript line 174 to 180.

2) As described in Sect 2.2.4, the authors has finally employed the new method for VCD estimation in the mobile measurements, which is recommended by Wagner et al., 2010. Maybe the authors could shorten the introduction of the geometric approximation and standard method for the VCD estimation. Alternatively, I suggest the authors to provide the comparison of the retrieved VCD results between standard and suggested methods? For example, taking one day as an example, to present the time series of the DSCD_mea, DSCD_offset and AMF_trop as used in the E.q. (12).

**Response:** We followed the reviewer suggestion and shortened the introduction of the geometric approximation and standard method for the VCD estimation. Changes are listed in the following:

a) Line 149-150: we put the formula ($SCD_{meas} = SCD_{trop} + SCD_{strat}$) into the sentence

b) Line 151-152: sentence combination: "it can be assumed that the light path in the stratosphere for zenith and off zenith measurements are very similar, i.e. $SCD_{strat}(\alpha) \approx SCD_{strat}(90°)$"

c) Line 157-158: add description of $VCD_{trop}$: "and the tropospheric vertical column density ($VCD_{trop}$) can be expressed as follows"

d) Line 193-195: remove this paragraph in the ACPD version

In addition, we have compared the VCD retrieved with both standard and suggested method. Time series of the DSCD_meas, DSCD_offset, AMF_trop and VCD_trop on 28 Nov 2015 are shown in Figure R1. Fig. R1a shows the time series of $NO_2$ DSCDs for both elevation angles. Higher DSCDs are obtained from the lower elevation angle. The offset caused by the $NO_2$ absorption in the Fraunhofer reference spectrum and the stratospheric absorption (see Eq. 7 in the revised manuscript) are determined by a polynomial fit. $DSCD_{offset(t)}$ is then calculated following Eq. 10. The tropospheric VCDs of $NO_2$ calculated with both approaches (corresponding to Eq. 7 and Eq. 8 in the revised manuscript) are shown in Fig. R1d. Comparing with standard method, the temporal development of tropospheric $NO_2$ VCDs calculated by the new method is less noisy. Similar results can be found for $SO_2$ and HCHO in Figs. R2-3.

[Figure]

**Figure R1.** An example of determination of tropospheric $NO_2$ from the spectra measured on 28

November 2015. (a) NO$_2$ DSCDs, (b) DSCD$_{offset}$ (see Eq. 10) plotted as a function of time (green points), (c) Tropospheric NO$_2$ AMFs calculated by the radiative transfer model SCIATRAN, (d) The tropospheric VCDs of NO$_2$ calculated by new and standard method (corresponding to Eq. 7 and Eq. 8).

[Figure]

**Figure R2.** Same as Fig. R1, but for an example of determination of tropospheric SO$_2$ from the spectra measured on 27 November 2015.

[Figure]

**Figure R3.** Same as Fig. R1, but for an example of determination of tropospheric HCHO from the spectra measured on 2 December 2015.

3) In this study, OMI VCDs are computed using atmospheric profiles from WRF-Chem simulations. It would be interesting to show how the VCDs are different from other operational products, e.g., NASA product.

**Response:** In this study, the USTC's OMI tropospheric $NO_2$ product is used. This product has been reported to be more suitable for atmospheric conditions in China (Liu et al., 2016;Xing et al., 2017;Su et al., 2017). Correlations of daily averaged tropospheric $NO_2$ VCDs measured by MAX-DOAS with USTC OMI and NASA OMI satellite data are shown in Fig. R4. Compared to the NASA's standard product, the USTC's OMI tropospheric $NO_2$ VCD agrees better with ground measurements a Pearson correlation coefficient (R) of 0.82 while the correlation between MAX-DOAS and the NASA OMI product is 0.76. The results suggested that accounting for the local atmospheric conditions and use the WRF-Chem model with measured climatology parameter and newest emission inventory to simulate trace gas profile in AMF calculation could improve the accuracy of OMI $NO_2$ VCD products. The Figure 7 is replaced by Figure R4 in the revised manuscript. These information are now supplemented in the manuscript line 332 to 336.

[Figure]

**Figure R4.** (a) Correlation analysis and (b) time series of tropospheric NO2 VCDs measured by ship-based MAX-DOAS and OMI during Yangtze River campaign. MAX-DOAS data (black markers) are temporally averaged around the USTC OMI and NASA OMI overpass time (red and blue markers, respectively), while the OMI data are spatially averaged within 20 km radius around the ship's averaged position. The error bars show the 1σ standard deviations of ship-based MAX-DOAS and OMI data.

4) It is not clear how the pollution events are identified. From Figure 3, the $NO_2$ VCD on 29 of

Nov is also very high, however, this day is not identified as pollution event. The authors should provide more information on the criterion in selecting pollution events.

**Response:** Pollution events were identified with both $NO_2$ and $SO_2$ VCDs reached or above the threshold value of $4.0 \times 10^{16}$ molec/cm$^2$. Although the $NO_2$ VCD on 29 November is also very high, but the $SO_2$ VCD is relatively low. Thus, 29 November is not included in the pollution events analysis. The threshold value for identifying the pollution events is supplemented in the manuscript line 278.

5) In section 3.1, trajectories are calculated to assess the pollutant transportation. However, it is not clear that these backward trajectories are calculated at which altitude level. It would be much better to show the height of the backward trajectories as well, so that readers could better interpret how pollutants are transported.

**Response:** Considering atmospheric pollutants are mainly concentrated in low altitudes during heavy pollution episodes, the trajectory arrival heights were set to 500 m and assumed to be representative for the entire boundary layer. In this study, 24 h back-trajectories were calculated by the HYSPLIT trajectory model. This information is now included in the manuscript line 302.

6) It's novel and interested to identify the industrial and vehicle contribution by E.q. 13 and 14, which provide the new insights for the sources appointment. As one of the most developed area in China, there were some more studies focusing on the emission inventories and source appointment for YRD areas. Maybe the authors can review previous results and compare with this study a little bit more in the discussion.

**Responses:** We have updated the section referring to a recent publication (Xia et al., 2016), which show a dramatic growth of the number of vehicle plays an increasingly significant role for regional $NO_2$ pollution over past years. For Hubei and Jiangxi provinces, the number of power plant is less than Jiangsu province. Therefore, the contribution from vehicle emission to ambient $NO_2$ level is expected to be more pronounced with the dramatic growth of vehicle number. Changes are applied to line 415 to 418.

7) For the estimation of primary and secondary sources of HCHO, I have two questions: 1) which

kind of the measured HCHO concentrations was used in the regression model? Since the HCHO levels were determined by MAX-DOAS as VCDs, however, the ambient HCHO concentrations were used in the model. How the authors obtain the ground surface concentration HCHO from the VCDs? Otherwise, why the authors can make regression of the HCHO VCDs with in-situ CO, Ox?

2) For the diurnal pattern, the authors inferred that secondary formation of HCHO shows a peak value during noon time (11:00-14:00), however, there was also another peak of relative contribution of secondary sources around 10:00 LT. How to consider this phenomenon?

**Response:** (1) For the estimation of primary and secondary sources of HCHO, the measured HCHO used in the regression model is ground mixing ratios. Usually, surface HCHO mixing ratios can be obtained from the HCHO vertical profiles (e.g., 0-200 m layer) (Wang et al., 2014). As the viewing elevation angles of the MAX-DOAS measurements only include 30° and 90°, therefore, there is not enough information to retrieve HCHO vertical profiles. In this study, we use a simplified formula introduced by (Lee et al., 2008) to convert mean HCHO DSCDs to mixing ratios (ppbv).

$$M(\text{ppb}v) = 1.25 \times \frac{DSCD(molecule\ cm^{-2})}{dAMF} \times \frac{1}{2.688 \times 10^{16}(molecule\ DU^{-1})} \times \frac{1}{\Delta P(atom)}$$

where M is the mixing ratio, DSCD is the difference between the SCDs of the measured spectrum and that of the Fraunhofer reference spectrum, dAMF is a differential air mass factor (dAMF=AMF(α=30°)-AMF(α=90°)), and ΔP is the pressure difference between surface and 500 m height of boundary layer. The AMFs for this study were calculated using the radiative transfer model SCIATRAN 2.2 as described in Section 2.2.4.

(2) We agree with the reviewer that there is another peak of relative contribution of secondary sources at around 10:00 LT. The 24 h backward trajectories on 3 December (Fig. 6d in the revised manuscript) suggested that rapid transport of air masses carries significant amount of pollutants including formaldehyde precursor from polluted area in northern China. Thus, the peak of relative contribution of secondary sources around 10:00 LT is probably resulted from the transportation of formaldehyde precursor. The explanation of the diurnal pattern has been modified in the revised manuscript (Line 474-479 in the revised manuscript).

**Technical corrections:**

1) Introduce abbreviation on the first time used in the manuscript, e.g., line 169 WRFChem.

**Response:** Abbreviation (WRF-Chem) is now introduced at the first time used in the manuscript (Line 170).

2) Text on Figure 3-6 is too small to read. Considering the similarities of Fig.3 to Fig.5, I suggest to merge them together to show the time series of these three pollutants in each panel with a continuous X-axis.

**Response:** We have merged Fig.3-5 so that the time series of three pollutants ($NO_2$, $SO_2$, and HCHO) in each panel with a continuous X-axis. Considering that a single plot covering all data is a bit difficult to read, so we have now separated it into 2 figures (Fig. 3&4 in the revised manuscript) for two periods (from Nov 22 to Nov 28 and from Nov 29 to Dec 4). In addition, we have enlarged the fonts in figure 6.

3) Put error bars on Figure 13 and 14.

**Response:** We have now included error bars in Figure 14 (Fig. 13 in the revised manuscript).

4) Please uniform the units of VCDs. It's different in Figures (molec/cm2) and manuscript text (molec cm-2) now.

**Response:** We have uniformed the units of VCDs (molec/$cm^2$) both in Figures and manuscript text.

Dec. 2 might be due to the strong influence by the aerosols. This is an interesting episode because it might show that the effect of neglecting aerosol in satellite AMF calculations on satellite $NO_2$ retrieval. Please prove this hypothesis using the Lidar measurement data.

**Response:** We agree with the reviewer suggestion and estimated the influence of ignoring aerosols in satellite AMF calculation on $NO_2$ retrieval. As lidar measurements only cover limited area, therefore, we have recalculated OMI $NO_2$ VCDs by taking aerosol information from WRF-Chem simulation into account. Figure R1a shows OMI $NO_2$ VCDs calculated without consider aerosol in the AMF calculation while Figure R1b shows the OMI $NO_2$ VCDs recalculated using aerosol information from WRF-Chem simulation. The spatial distributions of $NO_2$ are changed after including aerosol in the radiative transfer calculation. Significant enhancement of $NO_2$ VCDs can be observed over some areas. However, the spatial patterns of $NO_2$ detected by MAX-DOAS and OMI are still quite different. The result indicates the impact of aerosol could not fully explain the discrepancy between MAX-DOAS and OMI on this day. On the other hand, the MAX-DOAS and OMI observations agree better during OMI overpass time (black star symbols) and the agreement decay when the time differences between MAX-DOAS and OMI measurements become larger. This implies a strong temporal variability of $NO_2$ on this day. We have now supplemented the additional information and explanation in the manuscript line 352-360.

[Figure]

**Figure R1.** Spatial pattern of tropospheric $NO_2$ VCD measured by OMI calculated under two different AMFs.

10) In section 3.1, Line 307-308: "In contrast, lower HCHO VCDs were observed mainly on rainy,

cloudy and haze days". Could the authors explain more about the possible reasons for this phenomenon?

**Response:** Lower HCHO VCDs were observed mainly on rainy, cloudy and haze days, which might be due to stronger wet deposition and weaker solar irradiation during these days. The possible reasons for this phenomenon are supplemented in the revised manuscript line 316-318.

11) In section 3.4, please explain how to convert HCHO VCD to HCHO concentrations (ppb).

**Response:** For the estimation of primary and secondary sources of HCHO, the measured HCHO used in the regression model is ground mixing ratios. Usually, surface HCHO mixing ratios can be obtained from the HCHO vertical profiles (e.g., 0-200 m layer) (Wang et al., 2014). As the viewing elevation angles of the MAX-DOAS measurements only include 30° and 90°, therefore, there is not enough information to retrieve HCHO vertical profiles. In this study, we use a simplified formula introduced by (Lee et al., 2008) to convert mean HCHO DSCDs to mixing ratios (ppbv).

$$M(\text{ppb}v) = 1.25 \times \frac{DSCD(molecule\ cm^{-2})}{dAMF} \times \frac{1}{2.688 \times 10^{16}(molecule\ DU^{-1})} \times \frac{1}{\Delta P(atom)}$$

where M is the mixing ratio, DSCD is the difference between the SCDs of the measured spectrum and that of the Fraunhofer reference spectrum, dAMF is a differential air mass factor (dAMF=AMF($\alpha$=30°)-AMF($\alpha$=90°)), and $\Delta$P is the pressure difference between surface and 500m height of boundary layer. The AMFs for this study were calculated using the radiative transfer model SCIATRAN 2.2 as described in Section 2.2.4. These information are now supplemented in the revised manuscript line 433-434.

12) In section 3.4, Line 430: "As other factors can also affect the atmospheric HCHO concentration". Please describe the "other factors" in more detail.

**Response:** In addition to primary emission and secondary formation of HCHO, meteorological conditions, e.g., solar irradiance, could also affect the atmospheric HCHO concentration. This information is now included in the revised manuscript line 451.

**Technical corrections:**

1) Line 321: 'Dec 1 and 3'> 'Dec 2 and 4'

**Response:** Corrected.

2) Line 344: 'aerosol' > 'aerosols'

**Response:** Corrected.

3) Text on Figure 2: change "dSCD" to "DSCD", and please uniform the units of DSCD (molec/cm$^2$).

**Response:** We have changed "dSCD" to "DSCD" on Figure 2 and uniform the units of DSCD (molec/cm$^2$) in the revised manuscript.

4) Explain the caption 'RTM' in Figure 3-5. Or remove it.

**Response:** We have removed the caption 'RTM' in Figures 3-5. We have merged Figs.3-5 so that the time series of three pollutants ($NO_2$, $SO_2$, and HCHO) in each panel with a continuous X-axis (Fig. 3&4 in the revised manuscript).

5) Text on Figure 6 is too small to read. Please use larger font size and put the colorbar on the right.

**Response:** We have enlarged the fonts in Figure 6 and put the colorbar on the right (Fig. 5 in the revised manuscript).

6) Improve the quality of Figure 10 (too many stripes).

**Response:** We have do our best to improve the quality of Figure 10, but too many stripes in Fig. 10 is inevitable which caused by the different satellite orbit.